# Conditional ablation of heparan sulfate expression in stromal fibroblasts promotes tumor growth *in vivo*

Ayumi Niwa[1], Toshiaki Taniguchi[1], Hiroyuki Tomita[1]*, Hideshi Okada[2],
Takamasa Kinoshita[1,3], Chika Mizutani[4], Mikiko Matsuo[1], Yuko Imaizumi[1],
Takahito Kuroda[1], Koki Ichihashi[1], Takaaki Sugiyama[1], Tomohiro Kanayama[1],
Yu Yamaguchi[5], Shigeyuki Sugie[6], Nobuhisa Matsuhashi[4], Akira Hara[1]

1 Department of Tumor Pathology, Gifu University Graduate School of Medicine, Gifu City, Japan,
2 Department of Emergency and Disaster Medicine, Gifu University Graduate School of Medicine, Gifu City,
Japan, 3 Department of Neurosurgery, Gifu University Graduate School of Medicine, Gifu City, Japan,
4 Department of Gastroenterological Surgery/Pediatric Surgery, Gifu University Graduate School of
Medicine, Gifu City, Japan, 5 Sanford Burnham Prebys Medical Discovery Institute, La Jolla, California,
United States of America, 6 Department of Pathology, Asahi University Hospital, Gifu, Japan

☯ These authors contributed equally to this work.
* h_tomita@gifu-u.ac.jp

**Data Availability Statement:** All relevant data are within the paper and its Supporting Information files.

## Abstract

Heparan sulfate (HS) is a glycocalyx component present in the extracellular matrix and cell-surface HS proteoglycans (HSPGs). Although HSPGs are known to play functional roles in multiple aspects of tumor development and progression, the effect of HS expression in the tumor stroma on tumor growth *in vivo* remains unclear. We conditionally deleted *Ext1*, which encodes a glycosyltransferase essential for the biosynthesis of HS chains, using S100a4-Cre (*S100a4-Cre; Ext1^{f/f}*) to investigate the role of HS in cancer-associated fibroblasts, which is the main component of the tumor microenvironment. Subcutaneous transplantation experiments with murine MC38 colon cancer and Pan02 pancreatic cancer cells demonstrated substantially larger subcutaneous tumors in *S100a4-Cre; Ext1^{f/f}* mice. Additionally, the number of myofibroblasts observed in MC38 and Pan02 subcutaneous tumors of *S100a4-Cre; Ext1^{f/f}* mice decreased. Furthermore, the number of intratumoral macrophages decreased in MC38 subcutaneous tumors in *S100a4-Cre; Ext1^{f/f}* mice. Finally, the expression of matrix metalloproteinase-7 (MMP-7) markedly increased in Pan02 subcutaneous tumors in *S100a4-Cre; Ext1^{f/f}* mice, suggesting that it may contribute to rapid growth. Therefore, our study demonstrates that the tumor microenvironment with HS-reduced fibroblasts provides a favorable environment for tumor growth by affecting the function and properties of cancer-associated fibroblasts, macrophages, and cancer cells.

## Introduction

All cells of the human body are covered by a dense layer of sugars, proteins, and lipids, collectively termed the glycocalyx. Heparan sulfate (HS), a member of the glycosaminoglycan family,

**Funding:** This research was supported by JSPS KAKENHI under Grant Number 20K07587 (H.T.). The funders had no role in study design, data collection and analysis, decision to publish, or preparation of the manuscript.

**Competing interests:** The authors have declared that no competing interests exist.

is an abundant component of the glycocalyx [1]. HS covalently attaches to a core protein to form HS proteoglycans (HSPGs) [2]. HSPGs are involved in cell–cell and cell–extracellular matrix (ECM) interactions and act as co-receptors for various growth factors and chemokines [3–6]. Owing to their effects on cellular interactions and signaling, HSPGs play important roles in tissue development and homeostasis [7] and under pathological conditions, most notably, tumor development and progression by regulating coagulation, growth factor signaling, and cell adhesion, proliferation, and mobility [8].

HS is a linear polysaccharide that is composed of variably sulfated repeating disaccharide units [9,10]. HSPGs are present in the intercellular space of mammalian tissues and on the surface of virtually all mammalian cells [11]. HS chain elongation is mediated by the action of the exostosin protein family members: exostosin-1 (EXT1) and EXT2 [12]. The primary role of EXT1 in HS polymerization has been established in several previous studies [13,14]. For example, HS synthesis is inhibited in embryonic stem cells lacking Ext1 [13].

Crosstalk between tumor cells and stromal fibroblasts is involved in tumor progression. Disrupting the bulky glycocalyx enhances fibroblast motility and increases proliferation and collagen synthesis [15]. Recently, syndecan-1, a stromal-derived HSPG, has been demonstrated to play an important role in the signaling between stromal fibroblasts and tumor cells [16]. Changes in the sulfation patterns and chain lengths may alter the affinity of stromal-derived HS for growth factors and cytokines. The biological activity of these factors may be altered during the tumor onset and growth [17]. However, information on how the changes in HS in the stroma affect tumor growth *in vivo* remains limited.

In the tumor stroma, non-cancer cells and components, such as fibroblasts, macrophages, lymphocytes, myeloid-derived suppressor cells (MDSCs), ECM, and blood vessels, are found around tumor cells. In the tumor microenvironment (TME), heterogeneous populations of cells and components play various roles in tumorigenesis [18]. Fibroblasts constituting the tumor stroma are cancer-associated fibroblasts (CAFs), a main component of TME [19,20]. CAFs are a heterogeneous population that are thought to enable tumor invasion through the TME by remodeling the ECM structure and interacting with tumor cells and other cells via the secretion of growth factors, cytokines, and chemokines [19,20]. Although numerous studies have been conducted on CAF, the complex interplay of several factors has hindered its comprehensive understanding.

Most studies have focused on HS in tumor cells [2,21]. Additionally, several studies have analyzed the relationship between fibroblasts and HS, including Ext1. However, to the best of our knowledge, these are mainly derived from *in vitro* experiments [14,17,22,23]. Therefore, we investigated the effects of abnormalities in the HS of CAFs on their function and tumor growth *in vivo*.

## Materials and methods

### Ethics statement and study approval

This human retrospective study was approved by the Institutional Review Board of Gifu University Hospital (no. 2022–054). Written informed consent was obtained from all the patients. All murine experiments were performed in accordance with the guidelines of the Gifu University International Animal Care and Use Committee (no. 2019–081).

### Experimental animals

S100a4-Cre and *Rosa26-CAG-LSL-tdTomato* (Ai9, Jax#:007905) mice were purchased from Jackson Laboratories (Bar Harbor, ME, USA). Additionally, six-week-old C57/BL6/J mice were purchased from Charles River (Kanagawa, Japan). The mouse *Ext1$^{flox}$* allele has been

described previously [24]. Male and female mice were bred and maintained under specific pathogen-free conditions in isolated ventilated cages in an air-conditioned room ($23 \pm 2°C$) with a 12-h light/dark cycle. The mice had free access to food and tap water. All animal experiments and breeding were conducted in accordance with the Regulations for Animal Experiments of Gifu University.

## Cell lines and culture

MC38 colorectal cancer and Pan02 pancreatic cancer cells were obtained from the National Cancer Institute (Bethesda, MD, USA). The cells were cultured in Dulbecco's modified Eagle's medium (DMEM) supplemented with 10% fetal bovine serum and 0.1% penicillin–streptomycin solution. The cells were incubated at 37°C and 5% $CO_2$. All experiments were performed with 10 or fewer cell passages from the frozen stock.

## Subcutaneous tumor transplantation in mice

MC38 or Pan02 cells ($1.0 \times 10^6$) were suspended in 100 μL of phosphate-buffered saline (PBS) and injected subcutaneously into the flank of 8–16-week-old C57/BL6 mice. When performing the procedure, the mice were anesthetized through the administration of three types of mixed intraperitoneal anesthesia, as previously reported by Kawai et al. [25]. Subcutaneous (S.C.) tumor volumes were measured daily using digital calipers as per the following formula:

Tumor volume = (tumor length × tumor width × tumor height × π)/6.

Mice implanted with MC38 cells or Pan02 cells were sacrificed 25–30 days or 36–40 days after transplantation, respectively. Tumor growth, health, and behavior of mice were monitored daily throughout the experiments. In the event of rapid tumor growth or cachexia, the mice were euthanized.

## Tissue preparation and histological examination

The mice were deeply anesthetized via isoflurane inhalation and euthanized by cervical dislocation. The euthanasia by isoflurane inhalation is an excellent method that is aimed to decrease pain and distress in the model animal [26]. The skin or subcutaneous tumors were removed from each mouse and cut into pieces.

For paraffin sectioning, all tissues were fixed in neutral buffered 10% formalin for 24 h and embedded in paraffin. Paraffin blocks were sectioned at 3-μm thickness and used for hematoxylin and eosin staining, Alcian blue staining (pH = 1.0 or 2.5), or immunostaining.

For frozen sections, the thorax of the mice was opened after administering anesthesia and the inferior vena cava was incised. Perfusion washing was performed using a drip and infusion device with equal volumes of cold 0.1 M PBS and cold 4% paraformaldehyde (PFA) solution. The skin or subcutaneous tumors were then excised, fixed in 4% paraformaldehyde overnight at 4°C, incubated in a 30% sucrose solution for cryoprotection at 4°C for 2–3 days, embedded in the optimal cutting temperature (OCT) compound, and frozen in liquid nitrogen. They were cut at 5-μm thickness using a cryostat (Leica Microsystems, Wetzlar, Germany). Each section was stored at −80°C, and the frozen sections were used for fluorescence observation and immunostaining.

## Human samples and immunohistochemistry

Human tissue sections were obtained from patients who underwent surgical resection at the Gifu University Hospital (Gifu, Japan) between 2006 and 2016. Paraffin blocks of human tissues were cut at 3-μm thickness, and the serial sections were subjected to

immunohistochemistry for Ext1, αSMA, CD8α, and CD163. Deparaffinized sections were autoclaved and boiled for immunostaining in 0.015 M sodium citrate buffer solution (pH 6.0) or Tris EDTA buffer solution (pH 9.0) at 110˚C for 10 min for heat-induced antigen retrieval. To block the endogenous peroxidase activity, the sections were incubated in 3% $H_2O_2$ diluted with methanol for 10 min. Non-specific antibody binding was blocked with 2% normal bovine serum for 40 min. Then, the sections were incubated with rabbit anti-Ext1 (dilution 1:50, Abcam, Cambridge, UK, ab126305), rabbit anti-αSMA (dilution 1:200, Abcam, ab5694), rabbit anti-CD8α (dilution 1:200, CST, Danvers, MA, #85336), or rabbit anti-CD163 (dilution 1:250, Abcam, ab119996) overnight at 4˚C. Then, the sections were incubated with peroxidase-labeled anti-rabbit antibody (Histofine Simplestain Max PO (R); Nichirei, Tokyo, Japan) for 60 min at 37˚C. Immunoreaction was visualized using 3,3′-diaminobenzidine tetrahydrochloride (DAB, Sigma), and the sections were counterstained with hematoxylin. Cases of colon and pancreatic cancers were classified into three levels—weak (score 0), intermediate (score 1), and strong (score 2)—according to the intensity of EXT1 staining of stromal fibroblasts. Additionally, to evaluate the intensity of αSMA staining of stromal fibroblasts, the cases of colon and pancreatic cancers were classified into three levels: low, middle, and high.

## Immunohistochemistry and immunofluorescence of mouse tissue sample

Paraffin blocks of mouse tissues were cut at 3-μm thickness, and the serial sections were subjected to immunohistochemistry for Ki-67, cleaved caspase-3, a-smooth muscle actin (αSMA), F4/80, CD11c, CD8α, and MMP-7. For immunostaining, the deparaffinized sections were autoclaved and boiled in 0.015 M sodium citrate buffer solution (pH 6.0) or Tris EDTA buffer solution (pH 9.0) at 110˚C for 10 min for heat-induced antigen retrieval. To block the endogenous peroxidase activity, the sections were incubated in 3% $H_2O_2$ diluted with methanol for 10 min. Non-specific antibody binding was blocked with 2% normal bovine serum for 40 min. The sections were then incubated with rabbit anti-Ki67 (dilution 1:500, CST, #12202), rabbit anti-cleaved caspase-3 (dilution 1:100, CST, #9664), rabbit anti-αSMA (dilution 1:200, Abcam, ab5694), rabbit anti-F4/80 (dilution 1:500, CST, #70076), rabbit anti-CD11c (dilution 1:400, CST, #97585), rabbit anti-CD8α (dilution 1:500, CST, #98941), or rabbit anti-MMP-7 (dilution 1:50, CST, #3801) overnight at 4˚C. Later, the sections were incubated with peroxidase-labeled anti-rabbit antibody (Histofine Simplestain Mouse Max PO (R); Nichirei) for 60 min at 37˚C. Immunoreaction was visualized using DAB (Sigma), and the sections were counterstained with hematoxylin.

For fluorescent immunostaining, the frozen sections were cut coronally at 5-μm thickness using a cryostat (Leica). The sections were incubated with rabbit anti-Ext1 (dilution 1:100, Bioworld, BS6597), mouse anti-heparan sulfate for the 10E4 clone (dilution 1:100, amsbio, #370255), or rabbit anti-αSMA (dilution 1:100, Abcam, ab5694), or rabbit anti-Vimentin (dilution 1:100, CST, #5741) overnight at 4˚C; then incubated with Goat Anti-Rabbit IgG H&L (Alexa Fluor® 488, dilution 1:250, Abcam, ab150077) or Goat Anti-Mouse IgG H&L (Alexa Fluor® 488, dilution 1:250, Abcam, ab150117) for 60 min at 37˚C, and finally stained with DAPI (dilution 1:1000, WAKO). Heparan sulfate staining was performed with specific mouse IgG blocking reagents—Histofine MOUSESTAIN Kit (Nichirei).

## Double immunofluorescence of human or mouse tissue samples

After cutting coronally at 5-μm thickness, the sections of human paraffin blocks were incubated with two primary antibodies—rabbit anti-Ext1 (dilution 1:50, Abcam, ab126305) and mouse anti-Vimentin (dilution 1:50, Dako, M0725)—overnight at 4˚C. Then, the sections were incubated with Goat Anti-Rabbit IgG H&L (Alexa Fluor® 594, dilution 1:300, Abcam,

ab150084) and Goat Anti-Mouse IgG H&L (Alexa Fluor® 488, dilution 1:200, Abcam, ab150117), and finally stained with DAPI (dilution 1:1000, WAKO).

The sections of the mouse paraffin blocks were incubated with two primary antibodies, rabbit anti-Ext1 (dilution 1:100, Bioworld, BS6597), and chicken anti-Vimentin (dilution 1:100, Abcam, ab39376) overnight at 4˚C. Then, the sections were incubated with Goat Anti-Rabbit IgG H&L (Alexa Fluor® 594, dilution 1:300, Abcam, ab150084) and Goat Anti-Chicken IgY H&L (Alexa Fluor® 488, dilution 1:200, Abcam, ab150173), and finally stained with DAPI (dilution 1:1000, WAKO).

## Immunofluorescence of fibroblast culture cells

We performed a primary culture procedure to establish mouse fibroblast cultures. Mice ears were placed in a Petri dish, and the tissue was cut into small pieces using a scalpel. The tissue was then incubated with collagenase stock solution (collagenase, Sigma-Aldrich) in Hank's balanced salt solution (Invitrogen) at 37˚C for 25 min. The tube was then centrifuged at 1000 rpm for 5 min to remove the supernatant, and 1 mL of HBSS was added and centrifuged at 1000 rpm for 5 min again to remove the supernatant. Then, 0.05% trypsin was added and incubated at 37˚C for 20 min. The supernatant was removed by centrifugation at 1000 rpm for 5 min, and the pellet was resuspended in DMEM supplemented with 10% fetal bovine serum and 0.1% penicillin–streptomycin solution, and spread on a chamber slide system (Thermo Fisher Scientific) except for the larger tissue. After 24 h of incubation at 37˚C, the cells on the chamber slides were fixed in 4% PFA for 15 min. After washing three times with PBS, the cells were permeabilized by incubation in a 0.2% Tween solution with PBS for 15 min. The cells were then incubated with rabbit anti-Ext1 (dilution 1:100, Bioworld, BS6597) or mouse anti-heparan sulfate for the 10E4 clone (dilution 1:100, amsbio, #370255) overnight at 4˚C, incubated with Goat Anti-Rabbit IgG H&L (Alexa Fluor® 488, dilution 1:250, Abcam, ab150077) or Goat Anti-Mouse IgG H&L (Alexa Fluor® 488, dilution 1:250, Abcam, ab150117) for 60 min at 37˚C, and finally stained with DAPI (dilution 1:1000, WAKO). Heparan sulfate staining was performed with specific mouse IgG blocking reagents—Histofine MOUSESTAIN Kit (Nichirei).

## Preparation of samples and flow cytometry

For dissociation of the subcutaneously transplanted tumor into a single-cell suspension, we used the BD Horizon™ Dri Tumor & Tissue Dissociation Reagent (TTDR; Becton, Dickinson and Company BD Biosciences, USA) according to the manufacturer's instructions. Briefly, peri-tumor and intra-tumor tissues were minced, digested using TTDR, and strained through a 70-μm filter to produce a single-cell suspension. Next, the cells were pelleted by centrifugation at 1,000 rpm for 5 min. Then, the pellet of intra-tumor tissues was dissolved in 5 nM SYTOX™ Red Dead Cell Stain (Invitrogen) per mL of cell suspension for dead cell selection, and incubated at room temperature for 15 min in the dark. After isolation of a single-cell suspension, the samples of peri-tumor and intra-tumor tissues were run on a Beckman Coulter CytoFLEX™ (Beckman Coulter, Brea, CA, USA), and the data were processed using the FlowJo software (Tree Star Inc., Ashland, OR, USA).

## Microarray analysis

Total RNA was isolated from subcutaneously transplanted tumors using a Maxwell RSC simplyRNA Tissue Kit (Promega Corp., Fitchburg, WI, USA; cat. #AS1340). Using the Low Input Quick Amp Labeling Kit 1-color (Agilent Technologies, Santa Clara, CA, USA; #5190–2305), Cy3-labeled probes were prepared from the total RNA. Then, the probes were hybridized with

a microarray slide (SurePrint G3 Mouse GE 8 × 60 K Microarray; Agilent Technologies) at 65˚C for 17 h. After washing, the slide was scanned with a microarray scanner (ArrayScan, Agilent Technologies) to obtain the fluorescent signal of the probes and then digitized using the Feature Extraction software (Agilent). Finally, gene expression analysis was performed using the GeneSpring GX software (Agilent). Probe names of *Cd80*, *Cd86*, *Ifng*, *Tnf*, *Il1β*, and *Mmp7* are listed in S1 Table. Significant differences between tumors from WT and *Ext1* $^{flox/flox}$; *S100a4-Cre* mice were identified using moderated t-tests. For all analyses, statistical significance was set at P ≤ 0.05.

## Reverse transcription reaction and real-time RT-PCR

The total RNA extracted by the above method was used and cDNAs were synthesized using the SuperScrit III First-Strand Synthesis Kit (Life Technologies, Inc., USA) according to the manufacturer's instructions. Real-time RT-PCR was performed using THUNDERBIRD SYBR qPCR Mix (Toyobo, Osaka, Japan) on a Step One Plus Real-Time PCR system (Applied Biosystems, Foster City, CA). To analyze relative gene expression, the comparative Ct method was used and duplicate reactions were performed. The expression levels were normalized with the expression of beta-actin. Relative values of gene expression levels were calculated by setting the expression level of the sample with the lowest expression level as 1. All primers are listed in S2 Table.

## Calculating the image data

The positive cell count of immunohistochemistry was performed with a cell counting command of NIH Image J software. Measurement of the thickness of the αSMA-positive fibroblasts layer was also performed with Image J. Additionally, the fluorescent images were transformed to 8-bit type and thresholds were set. Then, the area of Tomato-positive cells, the area of Tomato- and αSMA-positive fibroblasts, and the fluorescence intensity value of HS staining were measured with Image J.

## Statistical analysis

The statistical details of the mouse experiments can be found in the figure legends. Statistical significance was determined using the unpaired t-test and Mann–Whitney test. All statistical analyses were performed using the GraphPad Prism 9 software (GraphPad Software Inc., USA).

## Results

### HS reduction in the fibroblast-specific Ext1 knockout mice

Stromal fibroblasts with hypomorphic mutation of the HS elongating enzyme EXT1 have a low heparan sulfate content *in vitro* [17,23]. Hence, to investigate how fibroblast HS affects the tumor stromal environment *in vivo*, we generated fibroblast-specific *Ext1* conditional knockout mice by crossing *Ext1*$^{flox/flox}$ with *S100a4-Cre/+* mice. S100a4-Cre mice were used to elucidate the effects of fibroblast-specific gene deletion under fibrotic and neoplastic conditions [27,28]. The generated mutant mice were referred to as *S100a4-Cre; Ext1*$^{f/f}$ mice. *Ext1*$^{flox/flox}$ or *Ext1*$^{flox/+}$ mice were used as controls.

To investigate *Ext1* and *HS* expression under the control of the *S100a4-Cre* promoter in *S100a4-Cre; Ext1*$^{f/f}$ mice, we generated *S100a4-Cre; Ext1*$^{f/f}$, *lsl-tdTomato*, and control (*S100a4-Cre*; *lsl-tdTomato*) mice. Using these mice, we performed immunofluorescence staining to examine *Ext1* and *HS* expression.

In the subcutaneous tissues of the skin, *Ext1* expression in tdTomato-expressing fibroblasts of *S100a4-Cre; Ext1$^{f/f}$; Lsl-tdTomato* mice was almost lost compared to the control (*S100a4-Cre; Lsl-tdTomato*) mice (Fig 1A). Ext1 expression was observed other than fibroblasts in the subcutaneous tissue, and it was thought to be primarily expressed by endothelial cells [29,30]. Some prior literature suggests that it may be the expression of immune cells [31,32]. *HS* expression in tdTomato-expressing fibroblasts of *S100a4-Cre; Ext1$^{f/f}$; Lsl-tdTomato* mice was lower than that in the control (*S100a4-Cre; Lsl-tdTomato*) mice (Fig 1B). Additionally, to confirm the evaluation of *Ext1* and *HS* expression, we isolated and cultured fibroblasts from *S100a4-Cre; Ext1$^{f/f}$; Lsl-tdTomato* and control mice for 1 day and examined the expression of EXT1 and HS using immunofluorescent staining. Similar to that in the skin tissue in the mice (Fig 1A and 1B), *Ext1* expression in tdTomato-expressing fibroblasts of *S100a4-Cre; Ext1$^{f/f}$; Lsl-tdTomato* mice was lost compared to the control (*S100a4-Cre; Lsl-tdTomato*) mice (S1A Fig). *HS* expression in tdTomato-expressing fibroblasts from the *S100a4-Cre; Ext1$^{f/f}$; Lsl-tdTomato* mice was lower than that in the control (*S100a4-Cre; Lsl-tdTomato*) mice (S1B Fig). These results indicated that the loss of *Ext1* in the fibroblasts reduced the biosynthesis of HS *in vivo*. This reduction, but not complete loss, of HS solely by the deletion of EXT1 is supported by previous studies [33,34].

To determine whether the decrease in HS in fibroblasts affected the distribution of stromal mucins, we performed Alcian blue staining, which visualizes acidic connective tissue mucins, on the skin of *S100a4-Cre; Ext1$^{f/f}$*, and control mice. Alcian blue pH 2.5 staining, which visualizes sulfated and carboxylated acid mucopolysaccharides and glycoproteins, showed a marked decrease in connective tissues of the skin in *S100a4-Cre; Ext1$^{f/f}$* mice compared to the control mice (Fig 1C). Furthermore, Alcian blue pH 1.0 staining, which only recognizes sulfated acid mucopolysaccharides, also showed reduced staining in the connective tissues of the skin in *S100a4-Cre; Ext1$^{f/f}$* mice compared to the control mice (Fig 1C). These results revealed that a decrease in *HS* expression in fibroblasts alters the stromal environment of the mucus.

Some reports indicated that S100a4 is also expressed in immune cells and macrophages [35–37]. We performed immunofluorescence staining to examine vimentin expression of tdTomato-expressing fibroblasts in the subcutaneous tissue of the skin in *S100a4-Cre; Ext1$^{f/f}$; Lsl-tdTomato* and control mice. Tomato-positive fibroblasts were confirmed to be vimentin-positive in both mice (S3A Fig). Additionally, we confirmed Ext1- and vimentin-double positive fibroblasts in control mice (S3B Fig). Furthermore, histological images of lymph nodes, spleen, and subcutaneous tissue of the skin were compared between control and *S100a4-Cre; Ext1$^{f/f}$* mice. In the lymph nodes, both control and *S100a4-Cre; Ext1$^{f/f}$* mice showed clear formation of lymph follicles and no apparent changes in the tissue architecture. No apparent changes were also observed in the spleen. In the subcutaneous tissue of the skin, there was no apparent difference in the infiltrating immune cells (S1C Fig).

## Fibroblast-specific loss of Ext1 with HS reduction accelerates colon and pancreatic tumor growth

To examine how the HS-reduced stroma of *S100a4-Cre; Ext1$^{f/f}$* mice affected tumor growth, we subcutaneously transplanted murine tumor cells into *S100a4-Cre; Ext1$^{f/f}$* and control mice (Fig 2A). For this experiment, we used two tumor cells: MC38 murine colon and Pan02 pancreatic cancer cells. MC38 and Pan02 subcutaneous tumors engrafted into *S100a4-Cre; Ext1$^{f/f}$* mice grew significantly faster and larger than those engrafted into the controls (Fig 2B).

The MC38 and Pan02 tumors showed no evident differences in tumor cell morphologies between *S100a4-Cre; Ext1$^{f/f}$* and control mice (Fig 2C and 2D). To analyze the proliferation and apoptosis of tumor cells, we performed immunostaining using Ki-67 and cleaved caspase

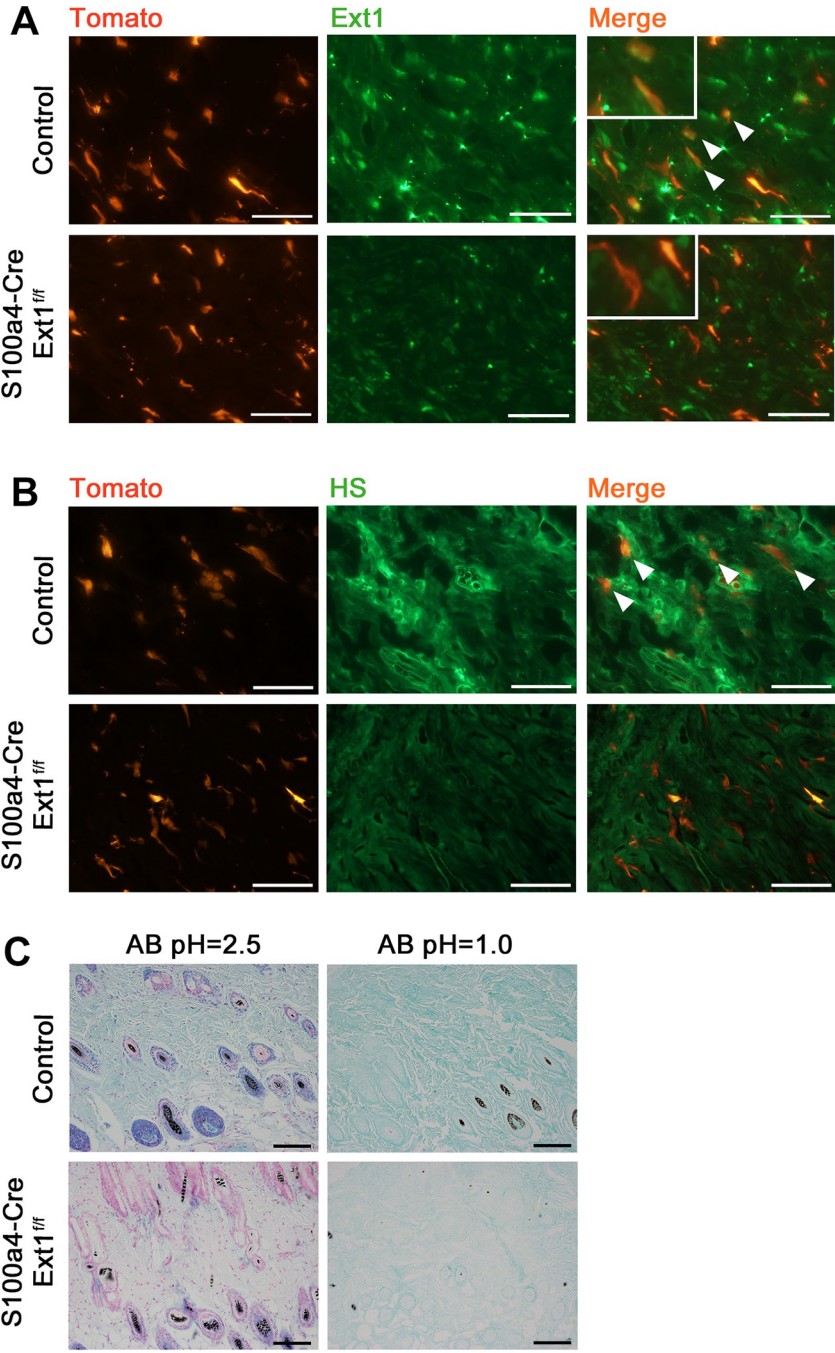

**Fig 1. Loss of exostosin-1 (EXT1), specific reduction of heparan sulfate (HS) in fibroblasts, and decreased acidic mucosubstances in the stroma of *S100a4-Cre; Ext1^{f/f}* mice.** (A) Immunostaining of Ext1 in subcutaneous tissues of skin of *S100a4-Cre; Ext1^{f/f}; Lsl-tdTomato* and control (*S100a4-Cre; Lsl-tdTomato*) mice. White arrowheads indicate Ext1-positive fibroblasts. Scale bar = 50 μm. (B) Immunostaining of HS in subcutaneous tissues of skin of *S100a4-Cre; Ext1^{f/f}; Lsl-tdTomato* and control (*S100a4-Cre; Lsl-tdTomato*) mice. White arrowheads indicate HS-positive fibroblasts. Scale bar = 50 μm. (C) Alcian blue staining (pH = 2.5 and 1.0) in subcutaneous tissues of skin of *S100a4-Cre; Ext1^{f/f}* and control (Ext1^{f/f}) mice. Scale bar = 100 μm.

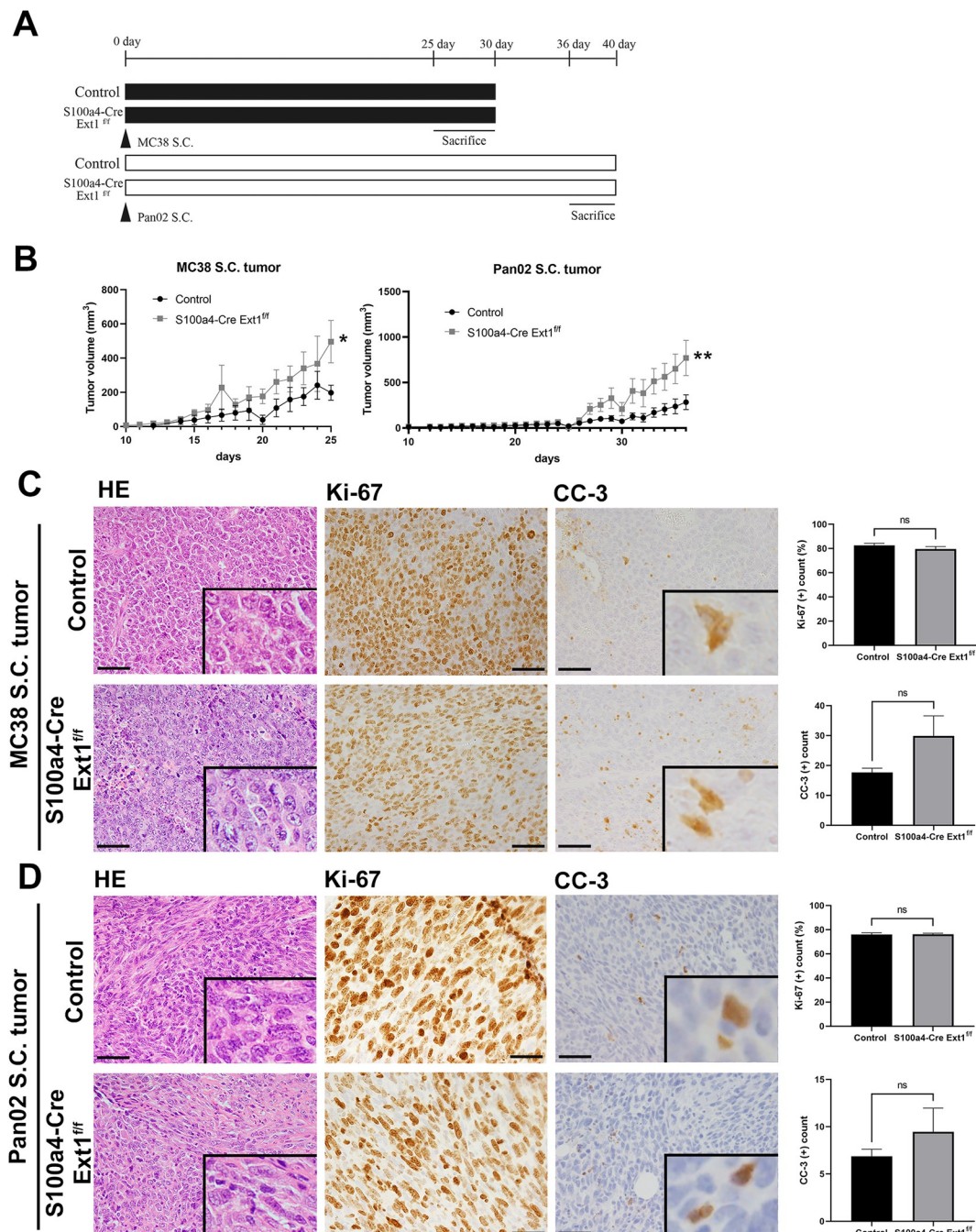

**Fig 2. *S100a4-Cre; Ext1^{f/f}* mice enhanced tumor growth in subcutaneous engrafted colon and pancreatic cancer cells.** (A) Experimental design of the subcutaneous tumor model. (B) The change of MC38 and Pan02 S.C. tumor volume in *S100a4-Cre; Ext1^{f/f}* and control mice. Each data point represents the mean tumor volume ± SEM. (MC38 S.C. tumor—control, N = 10, *S100a4-Cre; Ext1^{f/f}*, N = 16; Pan02 S.C. tumor—control, N = 12, *S100a4-Cre; Ext1^{f/f}*, N = 9; Mann–Whitney test, *P < 0.05, **P < 0.01). (C) Representative H&E staining of subcutaneous tumors and immunostaining of Ki-67 and cleaved caspase-3 (CC-3) of MC38 S.C. colon tumors in *S100a4-Cre; Ext1^{f/f}* and control mice (left). The ratio of Ki-67 and the number of CC-3-positive cells in tumors of *S100a4-Cre; Ext1^{f/f}* and control mice (right). Ki-67 data represent mean ± SEM, and CC-3 data represent mean ± SEM (N = 6 for each cohort, for both). Scale bar = 50 µm. (D) Representative H&E staining of subcutaneous tumors and immunostaining of Ki-67 and cleaved caspase-3 (CC-3) of Pan02 S.C. pancreatic tumors in *S100a4-Cre; Ext1^{f/f}* and control mice (left). The ratio of Ki-67 and the number of CC-3-positive cells in tumors of *S100a4-Cre; Ext1^{f/f}* and control mice (right). Ki-67 data represent mean ± SEM, and CC-3 data represent mean ± SEM (N = 6 for each cohort). Scale bar = 50 µm.

3 (CC-3) antibodies. There was no significant difference between *S100a4-Cre; Ext1^{f/f}* and control mice in MC38 and Pan02 tumor cells (Fig 2C and 2D) for Ki-67 immunostaining. For CC-3 immunostaining, there was a trend toward more positive cells in *S100a4-Cre; Ext1^{f/f}* compared to the controls in MC38 and Pan02 tumor cells, but the difference was not significant (Fig 2C and 2D).

These data indicate that HS reduction in fibroblasts of the surrounding environment enhances tumor growth and might be independent of the nature of the tumor cell itself.

## Myofibroblasts are reduced in the colon tumor of *S100a4-Cre; Ext1^{f/f}* mice

*Ext1* expression in fibroblasts is associated with αSMA-positive fibroblasts [38], also known as myofibroblasts and CAFs, in the TME. Additionally, HSPGs on the surface of stromal fibroblasts and tumor cells act as co-receptors in cytokine and growth factor signaling, thereby regulating cell proliferation, matrix production, cell migration, and the invasive properties of tumor cells [39].

Thus, to examine αSMA differentiation of fibroblasts in the TME, we performed immunostaining for αSMA antibody in MC38 S.C. colon tumor tissues of the *S100a4-Cre; Ext1^{f/f}* and control mice (Fig 3A). α-SMA (+) fibroblasts were found in the peri- and intra-tumor regions. In the peri-tumor region, we measured the thickness of the layer surrounding the αSMA (+) fibroblasts in the *S100a4-Cre; Ext1^{f/f}* and control mice. The layer of the MC38 S.C. tumors in the *S100a4-Cre; Ext1^{f/f}* mice was significantly thinner than that in the control mice (Fig 3A). Furthermore, Alcian blue staining was performed to compare the peri-tumor region, which was more intensely stained in the control mice than that in the *S100a4-Cre; Ext1^{f/f}* mice (Fig 3A).

In the intra-tumor region, to distinguish between αSMA (+) fibroblasts and vascular smooth muscle cells that express αSMA, we used *S100a4-Cre; Ext1^{f/f}; Lsl-tdTomato* and control (*S100a4-Cre; Lsl-tdTomato*) mice. First, we analyzed the area of Tomato-positive fibroblasts. Tomato-positive fibroblasts in the intra-tumor region were significantly reduced in the *S100a4-Cre; Ext1^{f/f}* mice compared to the control mice (Fig 3B). Additionally, we performed flow cytometry analysis on fresh tumor tissues to confirm the significant difference in Tomato-positive fibroblasts. The results showed that the percentage of Tomato-positive fibroblasts in peri- and intra-tumor regions of *S100a4-Cre; Ext1^{f/f}* mice was lower than that of control mice (Figs 3C and S2A).

To confirm the number of αSMA (+) fibroblasts in the intra-tumor region, we performed αSMA immunostaining in the tumor tissues of *S100a4-Cre; Ext1^{f/f}; Lsl-tdTomato* and control (*S100a4-Cre; Lsl-tdTomato*) mice. The ratio of myofibroblasts to Tomato-positive fibroblasts did not differ significantly between *S100a4-Cre; Ext1^{f/f}* and control mice; however, the ratio of the myofibroblast area to tumor area was significantly reduced in the *S100a4-Cre; Ext1^{f/f}* mice compared to the control mice (Fig 3D).

Furthermore, immunofluorescence staining of HS was performed on MC38 tumors of control and *S100a4-Cre; Ext1^{f/f}* mice. Compared to *S100a4-Cre; Ext1^{f/f}* mice, tumors of control mice showed more intense staining of HS, and the intensity value was significantly higher in control mice (S4A Fig).

We performed immunofluorescence staining to examine Ext1 and vimentin expression of Tomato-positive cells of MC38 tumor tissue in *S100a4-Cre; Ext1^{f/f}; Lsl-tdTomato* and control mice. Tomato-positive cells were confirmed to be Ext1-positive in control mice (S4B Fig), and Tomato-positive cells in both mice were confirmed vimentin-positive in both mice (S4C Fig).

These results showed that myofibroblasts decreased in the peri- and intra-tumor regions in the *S100a4-Cre; Ext1^{f/f}* mice compared to the control mice. Additionally, considering the

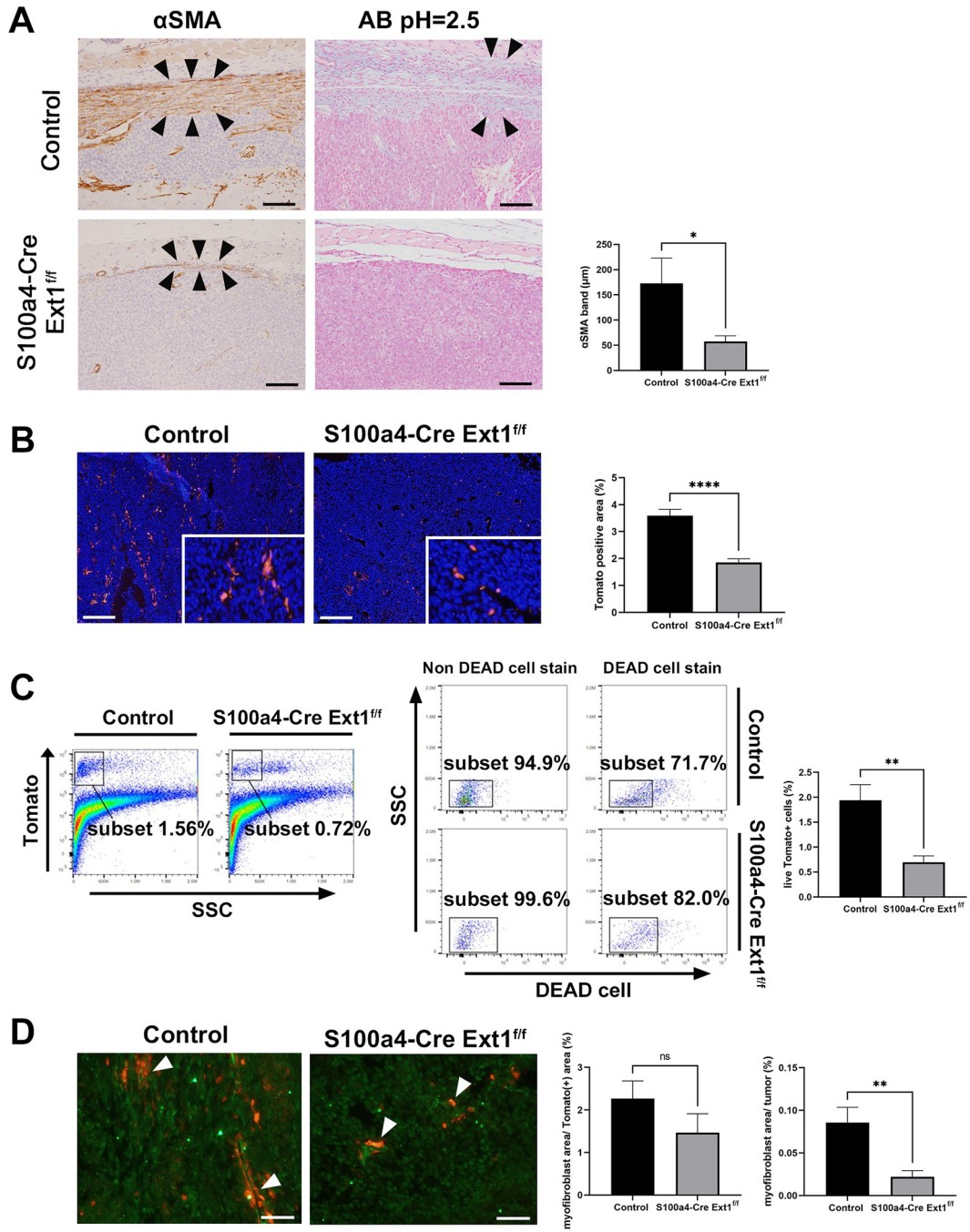

**Fig 3. Decreased peritumoral myofibroblasts in the MC38 S.C. tumors of *S100a4-Cre; Ext1^f/f* mice.** (A) Immunostaining of α-smooth muscle actin (αSMA)-positive spindle cells, i.e., myofibroblasts, in the peritumoral region of MC38 S.C. colon tumors in *S100a4-Cre; Ext1^f/f* and control mice (left). Arrowheads indicate the band comprising αSMA-positive cells. Alcian blue staining (pH = 2.5) in the peri-tumor region of MC38 S.C. colon tumors in *S100a4-Cre; Ext1^f/f* and control mice (middle). Arrowheads indicate the Alcian blue positive area. Measurement of the thickness of myofibroblasts around the MC38 S.C. tumor of *S100a4-Cre; Ext1^f/f* and control mice (right). Data represent mean ± SEM (N = 6 for each cohort, unpaired t-test, *P < 0.05). Scale bar = 100 μm. (B) Tomato expression in the intra-tumor region of MC38 S.C. tumors of *S100a4-Cre; Ext1^f/f; Lsl-tdTomato* and control (*S100a4-Cre; Lsl-tdTomato*) mice. Measurement of the area of Tomato-positive cells in both cohorts. Data represent mean ± SEM. N = 6 for each cohort unpaired t-test (*P < 0.05, **P < 0.01, ***P < 0.001, ****P < 0.0001). Scale bar = 100 μm. (C) Flow cytometric properties of intra-tumor region of MC38 S.C. tumor in *S100a4-Cre; Ext1^f/f; Lsl-tdTomato*, and control (*S100a4-Cre; Lsl-tdTomato)* mice (representative data) (left). The subset represents tdTomato-positive fibroblast cells (left). Dead cell selection in flow cytometry (representative data) (middle). Only living tdTomato-positive cells are collected and analyzed (middle). SSC, side scatter. The ratio of the number

of tdTomato-positive cells to total cells in the tumor (right). Data represent the mean ± SEM (N = 4 for each cohort, Mann–Whitney test, *$P < 0.05$, **$P < 0.01$). (D) Immunofluorescent staining of αSMA (green) in MC38 S.C. tumors of *S100a4-Cre; Ext1^{f/f}; Lsl-tdTomato* and control (*S100a4-Cre; Lsl-tdTomato*) mice (left). White arrowheads indicate αSMA-positive fibroblasts. Measurement of the percentage of the area of αSMA-positive fibroblasts, i.e., myofibroblasts, in MC38 S.C. tumors of *S100a4-Cre; Ext1^{f/f}; Lsl-tdTomato* and control (*S100a4-Cre; Lsl-tdTomato*) mice (right). The ratio of myofibroblast area to Tomato-positive fibroblast area and the ratio of myofibroblast area to the tumor area. Data represent the mean ± SEM (N = 6 for each cohort, unpaired t-test, *$P < 0.05$, **$P < 0.01$). Scale bar = 50 μm.

Alcian-blue stain and HS stain, HS production in fibroblast may itself be crucial for their infiltration in the MC38 tumor tissues.

Finally, to investigate the gene expression profile in the TME of MC38 S.C. tumors of *S100a4-Cre; Ext1^{f/f}* mice, we performed microarray analysis and gene set enrichment analyses (GSEA). The results revealed decreased expression of a gene set related to the TGF-β signaling pathway in MC38 S.C. tumors of the *S100a4-Cre; Ext1^{f/f}* compared to the control mice (S2B Fig). Furthermore, although no significant difference was observed, the *Tgfβ1* expression tended to be lower in the tumor of *S100a4-Cre; Ext1^{f/f}* mice (S6C Fig). Consequently, it was assumed that the decreased function of the TGF-β signaling pathway might be related to myofibroblast differentiation in the microenvironment of HS-reduced fibroblasts.

Tumor stromal fibroblasts, also defined as CAFs in the TME, are a heterogeneous population. αSMA-positive fibroblasts, which are myofibroblasts, are often reported to act in a tumor-promoting manner [40,41], but this remains a matter of debate. Our results suggest that a decrease in the overall number of fibroblasts and myofibroblasts within the tumor stroma, which comprises the HS-depleted stroma, may have enhanced tumor growth.

## Intra-tumor macrophages are reduced in the colon tumor of S100a4-Cre; Ext1^{f/f} mice

Several mechanisms are involved in the immunosuppressive capabilities of activated fibroblasts in the TME [42]. The most abundant cells in the TME are immune cells, such as macrophages, dendritic cells (DCs), and lymphocytes [43]. Thus, we focused on immune cells in the MC38 S.C. tumors of *S100a4-Cre; Ext1^{f/f}* and control mice. Immunostaining for F4/80, a macrophage marker, and for CD11c, a DC marker, was performed, and the number of intra-tumor positive cells was analyzed. The number of F4/80-positive cells in MC38 S.C. tumors of the *S100a4-Cre; Ext1^{f/f}* mice was significantly lower than that in the control mice (Fig 4A). There was no significant difference in the number of CD11c-positive cells of MC38 S.C. tumors between the *S100a4-Cre; Ext1^{f/f}* mice and the control mice. Next, to examine the difference in the number of cytotoxic T cells activated by antigen presentation, we performed immunostaining for CD8α. There were no significant differences in the number of cytotoxic T cells in the peri- and intra-tumor regions of *S100a4-Cre; Ext1^{f/f}* and control mice; however, there was a tendency for the number of CD8α-positive cells in the tumors of the *S100a4-Cre; Ext1^{f/f}* mice to be lower than that in the control mice (Fig 4B). These data suggest that colon tumor growth may be associated with a decrease in tumor-associated macrophages (TAM) in the TME with reduced αSMA(+) CAFs.

Additionally, we performed a microarray analysis of MC38 S.C. tumor tissues in *S100a4-Cre; Ext1^{f/f}* and control mice. The results showed that the RNA expression of *Cd80* and *Cd86*, factors related to antigen presentation, tended to decrease in the *S100a4-Cre; Ext1^{f/f}* mice (fold change >2; Fig 4C). By real-time RT-PCR analysis, we observed that the expression of *Cd80* and *Cd86* tended to be lower in the tumor of *S100a4-Cre; Ext1^{f/f}* mice and a significant difference in *Cd80* expression was observed (S6C Fig). CD80 and CD86 are members of the B7 superfamily that regulate T lymphocyte activation and tolerance, and primary antiviral CD8[+]

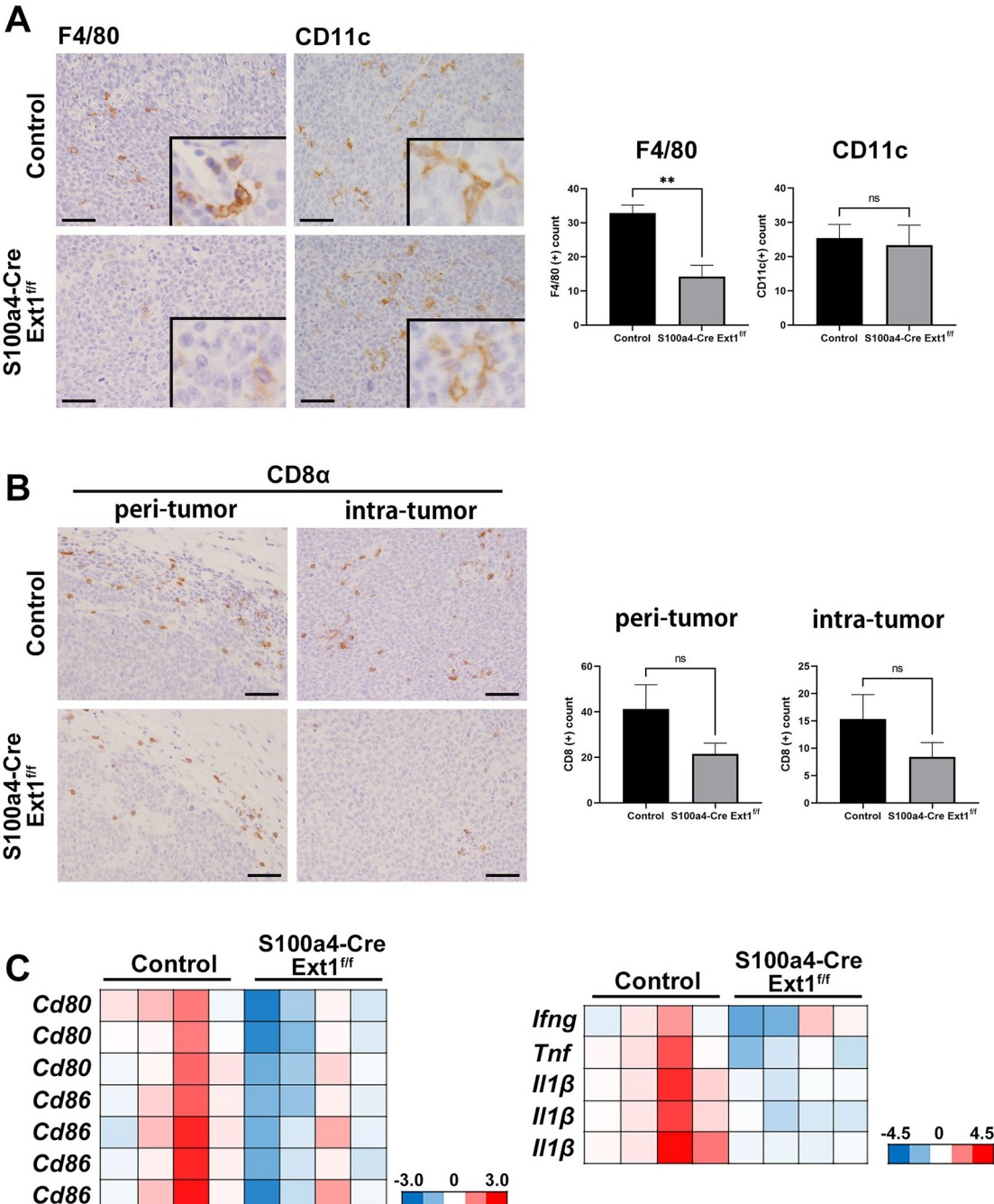

**Fig 4. Change of the immune microenvironment in MC38 S.C. tumors of *S100a4-Cre; Ext1^{f/f}* mice.** (A) Immunohistochemistry for F4/80 and CD11c of MC38 S.C. tumor of *S100a4-Cre; Ext1^{f/f}* and control mice (left). The number of F4/80- and CD11c-positive cells in tumors of *S100a4-Cre; Ext1^{f/f}* and control mice (right). Data represent mean ± SEM (N = 6 for each cohort, unpaired t-test, $^{*}P < 0.05$, $^{**}P < 0.01$). Scale bar = 50 μm. (B) Immunohistochemistry of CD8α in peri- and intra-tumor regions of MC38 S.C. tumors of *S100a4-Cre; Ext1^{f/f}* and control mice (left). The number of CD8α-positive cells in peri-and intra-tumor regions of tumors of *S100a4-Cre; Ext1^{f/f}* and control mice (right). Data represent mean ± SEM (N = 6 for each cohort, unpaired t-test). Scale bar = 50 μm. (C) Microarray analysis of MC38 S.C. tumor of *S100a4-Cre; Ext1^{f/f}* and control mice (N = 4 for each cohort). Decreased gene expression is indicated in blue and increased gene expression is indicated in red.

T cell responses are severely impaired in the absence of CD80/CD86 [44]. Similar to this anti-viral effect, our results suggest that a decrease in CD80/CD86 expression may be associated with CD8$^+$ T cell reduction in the tumor microenvironment of colon tumor growth of *S100a4-Cre; Ext1$^{f/f}$* mice.

Furthermore, by microarray analysis, the RNA expression levels of *Ifng*, *Tnf*, and *Il1β* decreased in the *S100a4-Cre; Ext1$^{f/f}$* mice (fold change >2) compared to those in the control mice (Fig 4C). By real-time RT-PCR analysis, the expression of *Ifng*, *Tnf*, and *Il1β* tended to be lower in the tumor of *S100a4-Cre; Ext1$^{f/f}$* mice, and there was significant difference in *Tnf* expression (S6C Fig). These data suggest that the TME with HS-reduced fibroblasts may be related to some immunosuppressive effects *in vivo*.

## Pancreatic tumor engrafted in *S100a4-Cre; Ext1$^{f/f}$* mice reduced peri-tumor myofibroblasts

Next, we analyzed fibroblasts in the Pan02 S.C. pancreatic tumors of *S100a4-Cre; Ext1$^{f/f}$* and control mice. In the peritumoral region, we measured the thickness of the layer surrounding the αSMA (+) fibroblasts in *S100a4-Cre; Ext1$^{f/f}$* and control mice. The layer of the Pan02 S.C. tumors in the *S100a4-Cre; Ext1$^{f/f}$* mice was significantly thinner than that in the control mice (Fig 5A), similar to MC38 S.C. colon tumors (Fig 3A). Unlike MC38 tumors, the intensity of Alcian blue staining in the peri-tumor region was not apparently different in *S100a4-Cre; Ext1$^{f/f}$* and control mice (Fig 5A).

To differentiate fibroblasts from vascular smooth muscle cells in the intra-tumor region, we used *S100a4-Cre; Ext1$^{f/f}$; Lsl-tdTomato* and control (*S100a4-Cre; Lsl-tdTomato)* mice, as in the MC38 colon tumor model (Fig 3B). First, we analyzed the area of Tomato-positive fibroblasts. The intratumoral region, unlike MC38 S.C. tumors, showed no difference in the total area of fibroblasts (Fig 5B).

To confirm the number of αSMA (+) fibroblasts in the intra-tumor region, we performed αSMA immunostaining in tumor tissues of *S100a4-Cre; Ext1$^{f/f}$; Lsl-tdTomato* and control (*S100a4-Cre; Lsl-tdTomato*) mice. The ratio of myofibroblasts to Tomato-positive fibroblasts did not differ significantly among the *S100a4-Cre; Ext1$^{f/f}$* and control mice (Fig 5C). Furthermore, the difference in the ratio of the myofibroblast area to tumor area was not significant among the *S100a4-Cre; Ext1$^{f/f}$* and control mice (Fig 5C).

Furthermore, immunofluorescent staining of HS was performed on Pan02 tumors of control and *S100a4-Cre; Ext1$^{f/f}$* mice. Compared to *S100a4-Cre; Ext1$^{f/f}$* mice, tumors in control mice showed more intense staining of HS, and the intensity value was significantly higher in control mice (S5A Fig).

We performed immunofluorescence staining to examine Ext1 and vimentin expression of Tomato-positive cells of Pan02 tumor tissue in *S100a4-Cre; Ext1$^{f/f}$; Lsl-tdTomato* and control mice. Tomato-positive cells were confirmed to be Ext1-positive in control mice (S5B Fig), and they were confirmed to be vimentin-positive in both mice (S5C Fig).

Altogether, unlike MC38 S.C. colon tumors, only peri-tumor myofibroblasts decreased in Pan02 S.C. tumors in the TME of the *S100a4-Cre; Ext1$^{f/f}$* mice compared to the control mice. Thus, to investigate the cause of the increase in tumor growth in Pan02 S.C. tumors in the *S100a4-Cre; Ext1$^{f/f}$* mice, the microarray data were analyzed. We found that *Mmp-7* RNA expression was significantly higher in the *S100a4-Cre; Ext1$^{f/f}$* mice than in the control mice (fold change >2, moderate t-test, $P < 0.05$; Fig 5D). By real-time RT-PCR analysis, the expression of *Mmp-7* was significantly higher in the tumor of *S100a4-Cre; Ext1$^{f/f}$* mice (S6D Fig).

To examine the number of Mmp-7-positive cells within the TME, immunostaining for Mmp-7 was performed and the number of Mmp-7-positive cells was measured. The results

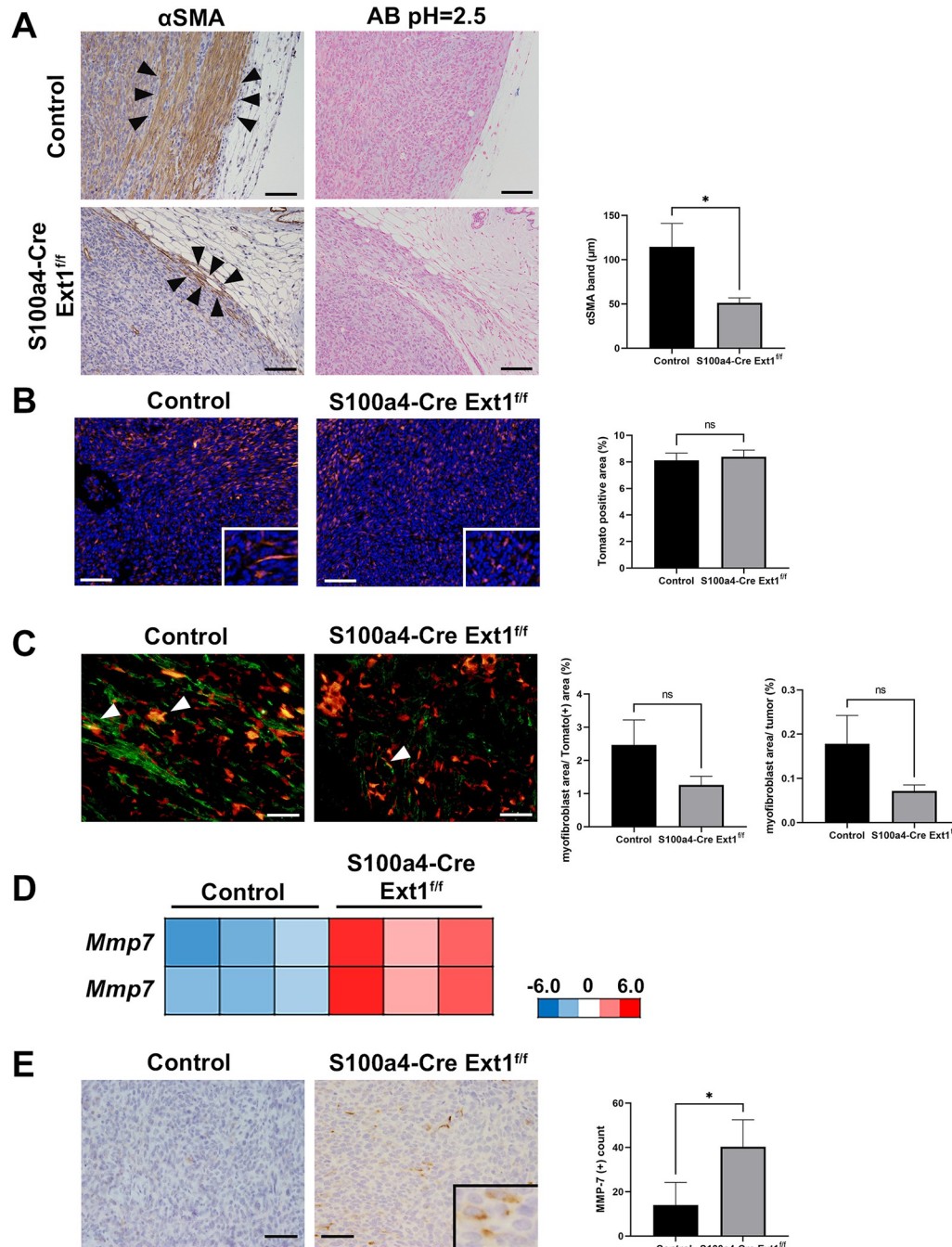

**Fig 5. Decreased myofibroblasts in the peritumoral region and high expression of MMP-7 in Pan02 S.C. tumor of** *S100a4-Cre; Ext1^{f/f}* **mice.** (A) Immunostaining of α-smooth muscle actin (αSMA)-positive spindle cells, i.e., myofibroblasts, in the peritumoral region of Pan02 S.C. pancreatic tumors in *S100a4-Cre; Ext1^{f/f}* and control mice (left). Arrowheads indicate the band comprising αSMA-positive cells. Alcian blue staining (pH = 2.5) in the peritumoral region of Pan02 S.C. pancreatic tumors in *S100a4-Cre; Ext1^{f/f}* and control mice (middle). Measurement of the thickness of myofibroblasts around the Pan02 S.C. tumor of *S100a4-Cre; Ext1^{f/f}* and control mice (right). Data represented as mean ± SEM (N = 6 for each cohort, unpaired t-test, $^*P < 0.05$). Scale bar = 100 μm. (B) Tomato expression in the intra-tumor region of Pan02 S.C. tumors of *S100a4-Cre; Ext1^{f/f}; Lsl-tdTomato* and control (*S100a4-Cre; Lsl-tdTomato*) mice. Measurement of the area of Tomato-positive cells in both cohorts. Data represented as mean ± SEM (N = 6 for each cohort, Mann–Whitney test). Scale bar = 100 μm. (C) Immunofluorescent staining of αSMA (green) in the Pan02 S.C. tumors of *S100a4-Cre; Ext1^{f/f}; Lsl-tdTomato* and control (*S100a4-Cre; Lsl-tdTomato*) mice (left). White arrowheads indicate αSMA-positive fibroblasts. Measurement of the percentage of the area of αSMA-positive fibroblasts, i.e., myofibroblasts, in Pan02 S.C. tumors of *S100a4-Cre; Ext1^{f/f}; Lsl-tdTomato* and control (*S100a4-Cre; Lsl-tdTomato*) mice

(right). The ratio of myofibroblast area to Tomato-positive fibroblast area and the ratio of myofibroblast area to tumor area were measured. Data represent mean ± SEM (N = 6 for each cohort, Mann–Whitney test, unpaired t-test, respectively). Scale bar = 50 μm. (D) Microarray analysis of Pan02 S.C. tumor of *S100a4-Cre; Ext1^{f/f}* and control mice (N = 4 for each cohort). Decreased gene expression is indicated in blue, and increased gene expression is indicated in red (fold change > 2, moderate t-test, *P* < 0.05; N = 3 for each cohort). (E) Immunostaining of MMP-7 in the Pan02 S.C. tumor of *S100a4-Cre; Ext1^{f/f}* and control mice (left). The number of MMP-7-positive cells in the tumors was measured (right). Data represent mean ± SEM (N = 6 for each cohort, Mann–Whitney test, *P* < 0.05).

showed a significant increase in the number of Mmp-7-positive cells in the *S100a4-Cre; Ext1^{f/f}* mice compared to the control mice (Fig 5E). MMP-7 is an important regulator of tumor formation in pancreatic cancer, and elevated MMP-7 levels correlate with metastasis and/or survival [45–47]. Thus, the increased Mmp-7 expression in our data might be associated with the rapid growth of pancreatic tumors in *S100a4-Cre; Ext1^{f/f}* mice.

Additionally, we performed immunostaining for F4/80 and for CD8α. In Pan02 tumors, there was no significant difference in the number of F4/80-positive cells and peri-tumor and intra-tumor CD8α-positive cells between control and *S100a4-Cre; Ext1^{f/f}* mice (S6A and S6B Fig).

Pan02 tumors were also analyzed for the differences in TGF-β signaling function. Unlike MC38 tumors, GSEA analysis did not show differences in the expression of gene set related to the TGF-β signaling pathway. However, real-time RT-PCR analysis showed significantly higher expression of *Tgfβ1* in the tumor of *S100a4-Cre; Ext1^{f/f}* mice (S6D Fig). Although there was a decrease in the myofibroblast of the intra-tumor region with MC38 tumor of *S100a4-Cre; Ext1^{f/f}* mice, there was no significant difference in Pan02, which may also be related to the difference in *Tgfβ1* expression between MC38 and Pan02.

## Evaluation of EXT1 expression in the fibroblasts of human colon and pancreatic cancers

To examine EXT1 expression in fibroblasts in the stroma of human colon and pancreatic cancers, we collected surgical specimens and performed immunostaining for EXT1 in both cohorts (colon cancer, n = 42; pancreatic cancer, n = 48).

In human colon and pancreatic cancers, we carefully evaluated EXT1 expression only in fibroblasts in the TME (Fig 6A and 6B). In human colon cancer, 12 (28.6%), 27 (64.3%), and 3 (7.1%) of 42 cases had the scores of 0, 1, and 2, respectively (Fig 6A). In human pancreatic cancer, 25 (52.1%), 20 (41.7%), and 3 (6.3%) of 48 cases had the scores of 0, 1, and 2, respectively (Fig 6B). The scoring results showed that human colon cancer had the highest number of score-1 cases, and human pancreatic cancer had the highest number of score-0 cases. Fewer cases of human colon and pancreatic cancers had a score of 2 (Fig 6A and 6B). Furthermore, we performed double immunofluorescence of Ext1 and vimentin on human cancer stroma, and we confirmed that Ext1-positive fibroblasts were vimentin-positive (S7A Fig).

These results indicate that EXT1 expression levels in fibroblasts of the TME vary, and specifically, lower levels of EXT1 are found in more than half of the human colon and pancreatic cancers.

Additionally, we performed αSMA staining and compared the cases of the each EXT1 score. For evaluating the expression of αSMA, because it was difficult to evaluate by number, the expression intensity was divided into three levels: low, middle, and high. In human colon cancer, score-0 cases (N = 3) comprised 1 low and 2 middle intensity cases, score-1 cases (N = 3) comprised 2 middle and 1 high intensity cases, and score-2 cases (N = 3) comprised 1 middle and 2 high intensity cases (S7B Fig). In pancreatic cancer, score-0 cases (N = 3) comprised 1 low, 1 middle, and 1 high intensity cases; score-1 cases (N = 3) comprised 2 middle

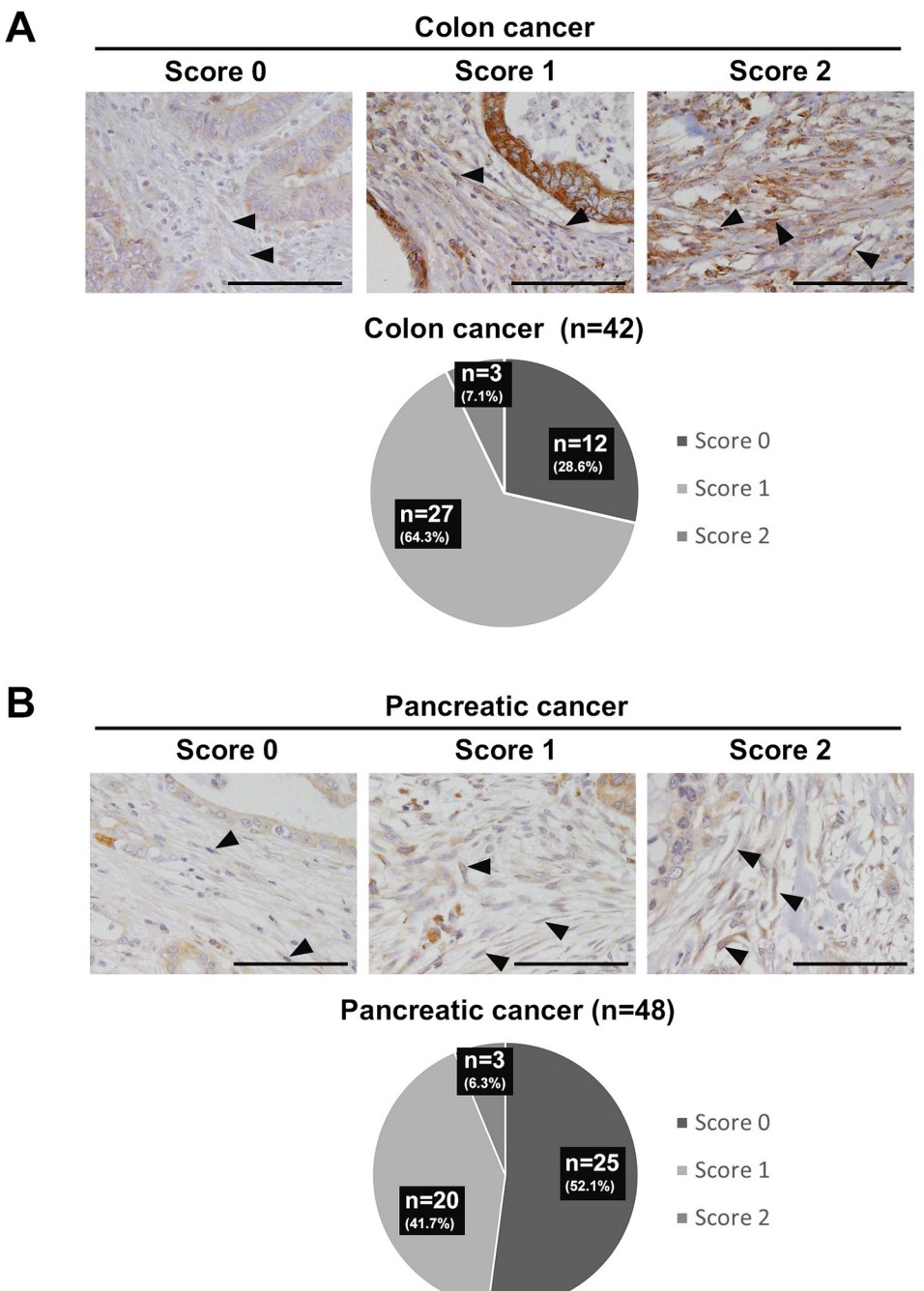

**Fig 6. Fibroblast-specific exostosin-1 (EXT1) expression in the stroma of human colon and pancreatic cancers.**
(A) Scoring of human colon cancers (N = 42). Arrowheads indicate Ext1-positive fibroblasts. Scale bar = 50 μm. (B) Scoring of human pancreatic cancers (N = 48). Arrowheads indicate Ext1-positive fibroblasts. Scale bar = 50 μm.

and 1 high intensity cases, and score-2 cases (N = 3) comprised 2 middle and 1 high intensity cases (S7B Fig).

Furthermore, we performed immunostaining for CD163, a human macrophage marker, and for CD8α, a cytotoxic T cells marker, and the number of intra-tumor positive cells was analyzed. No significant differences were observed in the number of CD163-positive cells

among the each EXT1 score cases of both colon and pancreatic cancers (S7C Fig). There was no significant difference in the number of CD8α-positive cells among each EXT1 score case of colon cancers. In pancreatic cancer, the number of CD8α-positive cells tended to be higher in EXT1 score-1 and EXT1 score-2 cases than those in EXT1 score-0 cases, and there was a significant difference between EXT1 score-0 and EXT1 score-1 cases (S7C Fig).

These results indicate that colon cancers with higher EXT1 scores tended to have higher αSMA staining intensity. In pancreatic cancer, low intensity was observed only in EXT1 score-0 cases, but there was no difference in αSMA staining intensity between EXT1 score-1 and EXT1 score-2 cases. A correlation between EXT1 and αSMA expression was observed more in colon cancer than in pancreatic cancer.

In pancreatic cancer, the number of CD8α-positive cells tended to increase with the increase in the score, with significant differences observed between EXT1 score-0 and EXT1 score-1 cases. Although no significant difference was observed between each scores in colon cancer, it may be suggested that the expression of EXT1 in the stroma is associated with the number of CD8α-positive cells in human cancers. However, due to the small number of cases, further evaluation is needed to examine the correlation of the expression of EXT1 and αSMA, and EXT1 and the number of CD8α-positive cells.

## Discussion

In this study, we revealed that stroma with HS-reduced fibroblasts decreased the cancer-associated myofibroblasts in the TME and promoted tumor growth *in vivo*. Furthermore, individual variability in the expression levels of EXT1 was observed in the CAFs of human cancer specimens.

EXT1 and EXT2 encode glycosyltransferases involved in the chain elongation step of HS biosynthesis [33]. Previously, it has been reported that *Ext1*-mutated (*Ext1*$^{Gt/Gt}$) fibroblasts exhibit short sulfated HS chains [14]. Additionally, the relationship between cell surface HS reduction, related to *Ext1* gene mutations, and fibroblasts has been reported using a spheroid tumor cell/fibroblast co-culture model [17]. In this co-culture model, HS-reduced *Ext1*$^{Gt/Gt}$ fibroblasts formed looser cell-matrix contacts and impaired tumor cell proliferation and migration compared to control wild-type fibroblasts. However, in contrast to the *in vitro* model, the *S100a4-Cre; Ext1*$^{f/f}$ mice in our *in vivo* model did not show impaired tumor cell proliferation in the colon and pancreatic cancers. Furthermore, *S100a4-Cre; Ext1*$^{f/f}$ mice showed a significant decrease in the surrounding myofibroblasts, the presence of which results in tissue stiffening [48], compared to control mice. Together, in a microenvironment with HS-reduced fibroblasts, the difficulty of fibroblasts to differentiate into myofibroblasts has been suggested to impact tumor growth significantly.

In colon tumors of the *S100a4-Cre; Ext1*$^{f/f}$ mice, the number of F4/80-macrophages was significantly lower than that in the control mice. Other immune cells, such as CD11c- DC and CD8-cytotoxic T cells, tended to decrease in the colon tumors of *S100a4-Cre; Ext1*$^{f/f}$ mice. Additionally, by microarray analysis, several inflammatory cytokines, such as *Infγ*, *Tnf*, and *Il1β* were downregulated in the TME, including colon tumor cells, immune cells, fibroblasts, and other cells, in the *S100a4-Cre; Ext1*$^{f/f}$ mice compared to those in the control mice. Through real-time PCR analysis, we found that the expression of *Infγ*, *Tnf*, and *Il1β* tended to be lower and the *Tnf* expression level was significantly lower in the *S100a4-Cre; Ext1*$^{f/f}$ mice compared to the control mice. These data suggest that at least HS-reduced fibroblasts in the TME can change the immune microenvironment. Consistent with our suggestion; Bao et al. [29] reported that EXT1 expression is important in immune responses since *Ext1* deficiency in endothelial cells in mice has been shown to impair lymphocyte recruitment in a contact hypersensitivity model because of impaired chemokine presentation. Owing to HS-reduced cells,

which constitute the immune microenvironment, these immunosuppressive microenvironments may promote immune tolerance and evasion through various mechanisms, although further studies are needed to verify this.

The TME is composed of various cell types, but fibroblasts and macrophages are recognized as the key cellular components of this niche [49]. Specifically, reciprocal interactions between macrophages and fibroblasts in the TME may be strong. Macrophages in the TME are generally referred to as tumor-associated macrophages (TAMs). TAMs can be pro- or anti-tumorigenic for reasons including ontogeny, location, or diversified functional subsets within the TME [49]. Furthermore, multiple factors contribute to phenotypic and functional heterogeneity. In our study, the underlying mechanisms remain unclear, but TAMs may be closely associated with CAFs.

Several studies have demonstrated that αSMA+ CAF depletion led to the increase in colorectal or pancreatic cancer progression using αSMA-tk transgenic mice, thereby leading to specific ablation of proliferating αSMA+ cells [50,51]. Our experiments also showed a decrease of αSMA+ CAF in MC38 colon tumors and Pan02 pancreatic tumors.

McAndrews et al. [50] showed that ablation of αSMA+ CAF in colorectal cancer leads to immunosuppression, including decreased CD8-positive T cells and increased Tregs. Although the difference was not significant, our study also showed that the number of CD8α-positive cells tended to decrease in *S100a4-Cre; Ext1^{f/f}* mice, and the decreased number of macrophages and the expression of some cytokines such as *Tnf* suggested that immunosuppression occurred in the *S100a4-Cre; Ext1^{f/f}* mice. McAndrews et al. [50] also showed that depletion of f αSMA + CAF increased the *Tgfβ1* secretion from the stromal cells. We observed the downregulation of the TGF-*β* signaling pathway in tumors of *S100a4-Cre; Ext1^{f/f}* mice in our study. Analysis of TGF-*β* expression throughout the tumor in our study may have made a difference.

Ozdemir et al. [51] showed that pancreatic cancer with αSMA+ CAF depletion showed a different immune milieu, with decreased CD8-positive T cells and increased Tregs. In our experiment, there was no significant difference in CD8α-positive T cells between *S100a4-Cre; Ext1^{f/f}* and control mice.

These differences may be because αSMA+ CAFs are not completely eliminated in our mouse model and that we analyzed tumors implanted subcutaneously. However, it can be said that the impact of αSMA+ CAF on tumor growth is significant.

MMP-7 expression was significantly increased in the pancreatic tumors of *S100a4-Cre; Ext1^{f/f}* mice. MMP-7 is abundantly expressed in numerous types of cancer tumor cells and overexpressed in precancerous cells and lesions [52,53]. MMP-7 promotes tumor progression by inhibiting apoptosis of cancer cells [54], reducing cell adhesion [55], and inducing angiogenesis [56]. Thus, MMP-7 can function as an oncogenic protein that regulates the occurrence and development of various tumors [57]. Numerous studies have showed that MMP-7 expression is elevated in pancreatic cancer compared to that in patients with pancreatitis and healthy individuals, and many studies have correlated MMP-7 expression levels with cancer metastasis and survival [58]. Fukuda et al. showed that in a mouse model of pancreatic cancer with KRAS mutations, loss of MMP-7 significantly reduced the tumor size and metastasis [47]. MMP-7 is assumed to be an important factor in the growth of pancreatic cancer, and the increased expression of MMP-7 in *S100a4-Cre; Ext1^{f/f}* mice may have led to increased tumor growth. Yu et al. reported that MMP-7 is stored in the extracellular stroma by binding to HS [59]. Although our experiments did not clarify the localization of MMP-7 and HS, the reduction of HS specifically upregulates MMP-7 expression in the pancreatic cancer stroma.

Germline heterozygous loss-of-function in Ext1 or Ext2 is a known cause of multiple exostoses (HME) [60,61]. Ext1 and Ext2 mutations have been reported in tumors other than HME. In leukemia and non-melanoma skin cancer cells, epigenetic inactivation of EXT1 by

promoter hypermethylation is found and the tumor suppressing effect of Ext1 was revealed [62]. However, in multiple myeloma, the correlation of high expression of EXT1 and a poor prognosis was reported [32]. In another previous paper, EXT2 mutation in breast carcinoma patients was reported [63]. Furthermore, Drake et al. [64] reported that both EXT1 and EXT2 have altered N-glycosylation in human aggressive breast cancer cells. In our study, the TME with Ext1 loss of fibroblasts proliferates colon and pancreatic cancer growth. Although our study did not analyze the expression of Ext1 and Ext2 in cancers in detail, it is suggested that changes in the expression of Ext1 or Ext2 in cancers may contribute to the differences in cancer development in this study. Further analysis of the expression of Ext1 and Ext2 in cancer cells and fibroblasts is needed to gain a better understanding of how TME with Ext1 loss of fibroblasts can lead to cancer development.

In this study, the tumor cells were transplanted subcutaneously, and this likely led to differences in the microenvironment compared to naturally occurring tumors. Furthermore, this study did not confirm the interaction between tumor cells and fibroblasts *in vitro*. Further analysis is needed to understand how HS reduction of fibroblasts enhances tumor growth *in vivo* because of the complexity of the analysis of the mechanisms and the heterogeneity of component cells in the TME. Furthermore, although we showed that S100a4 positive cells are fibroblasts with vimentin staining and histological images, some reports indicated that S100a4 is also expressed in immune cells and macrophages [35–37].

In conclusion, a TME with HS-reduced fibroblasts is favorable for cancer cell growth. Furthermore, altering the glycocalyx in the stroma affects the cancer cells and the associated immune system.

## Supporting information

**S1 Fig. Loss of exostosin-1 (Ext1) and specific reduction of HS in fibroblasts in *S100a4-Cre; Ext1^{f/f}* mice.** (A). Immunostaining for Ext1 in fibroblasts isolated from *S100a4-Cre; Ext1^{f/f}; Lsl-tdTomato* and control (*S100a4-Cre;Lsl-tdTomato*) mice. Ext1 expression was reduced in fibroblasts of *S100a4-Cre; Ext1^{f/f}* mice. White arrowheads indicate Ext1-positive fibroblasts. Scale bar = 50 μm. (B) Immunostaining for heparan sulfate (HS) in fibroblasts isolated from *S100a4-Cre; Ext1f/f, Lsl-tdTomato*, and control (*S100a4-Cre;Lsl-tdTomato*) mice. In control mice, the fibroblasts showed characteristic dot-like positive staining. White arrowheads indicate HS-positive fibroblasts. Scale bar = 50 μm. (C) Representative H&E staining of lymph node, spleen, and subcutaneous tissue of the skin in *S100a4-Cre; Ext1^{f/f}* and control mice. Scale bar = 50 μm.
(TIF)

**S2 Fig. MC38 S.C. colon tumor in *S100a4-Cre; Ext1^{f/f}* mice showed a decrease in the peri-tumor fibroblasts and expression of the gene set related to TGF-β signaling.** (A) Flow cytometric properties of the peri-tumor region of MC38 S.C. tumor in *S100a4-Cre; Ext1^{f/f}; Lsl-tdTomato*, and control (*S100a4-Cre; Lsl-tdTomato)* mice (representative data). The subset represents Tomato-positive fibroblast cells. SSC, side scatter. (N = 3 for each cohort) (B) Gene set enrichment analysis (GSEA) obtained from microarray analysis of MC38 S.C. tumor of *S100a4-Cre; Ext1^{f/f}* and control mice. MC38 S.C. tumors in *S100a4-Cre; Ext1^{f/f}* mice showed decreased expression of a set of genes related to TGF-β signaling compared to control mice. WT means control mice, and Mut means *S100a4-Cre; Ext1^{f/f}* mice (N = 4 for each cohort).
(TIF)

**S3 Fig. Tomato-positive fibroblasts and Ext1-positive fibroblasts showed vimentin-positive by immunostaining.** (A) Immunostaining of vimentin in subcutaneous tissues of the skin

of *S100a4-Cre; Ext1^{f/f}, Lsl-tdTomato*, and control (*S100a4-Cre; Lsl-tdTomato*) mice. White arrowheads indicate vimentin-positive fibroblasts. Scale bar = 50 μm. (B) Double immunofluorescence of Ext1 and vimentin in subcutaneous tissues of the skin of *S100a4-Cre; Ext1^{f/f}* and control (*Ext1^{f/f}*) mice. White arrowheads indicate Ext1 and vimentin double positive fibroblasts. Scale bar = 50 μm.
(TIF)

**S4 Fig. Immunostaining of HS, Ext1, and vimentin in MC38 S.C. colon tumor.** (A) Immunostaining of HS in MC38 S.C. tumor of *S100a4-Cre; Ext1^{f/f}; Lsl-tdTomato*, and control (*S100a4-Cre; Lsl-tdTomato*) mice (left). The intensity value of HS stain in tumors of *S100a4-Cre; Ext1^{f/f}; Lsl-tdTomato*, and control (*S100a4-Cre; Lsl-tdTomato*) mice (*right*). Data represent mean ± SEM. (N = 4 for each cohort, Mann–Whitney test, $^{*}P < 0.05$). Scale bar = 50 μm. (B) Immunostaining of Ext1 in MC38 S.C. tumor of *S100a4-Cre; Ext1^{f/f}; Lsl-tdTomato*, and control (*S100a4-Cre; Lsl-tdTomato*) mice. White arrowheads indicate Ext1-positive fibroblasts. Scale bar = 50 μm. (C) Immunostaining of vimentin in MC38 S.C. tumor of *S100a4-Cre; Ext1^{f/f}; Lsl-tdTomato*, and control (*S100a4-Cre; Lsl-tdTomato*) mice. White arrowheads indicate vimentin-positive fibroblasts. Scale bar = 50 μm.
(TIF)

**S5 Fig. Immunostaining of HS, Ext1, and vimentin in Pan02 S.C. pancreatic tumor.** (A) Immunostaining of HS in Pan02 S.C. tumor of *S100a4-Cre; Ext1^{f/f}; Lsl-tdTomato*, and control (*S100a4-Cre; Lsl-tdTomato*) mice (left). The intensity value of HS stain in tumors of *S100a4-Cre; Ext1^{f/f}; Lsl-tdTomato*, and control (*S100a4-Cre; Lsl-tdTomato*) mice (right). Data represent mean ± SEM. (N = 4 for each cohort, Mann–Whitney test, $^{*}P < 0.05$). Scale bar = 50 μm. (B) Immunostaining of Ext1 in Pan02 S.C. tumor of *S100a4-Cre; Ext1^{f/f}; Lsl-tdTomato*, and control (*S100a4-Cre; Lsl-tdTomato*) mice. White arrowheads indicate Ext1-positive fibroblasts. Scale bar = 50 μm. (C) Immunostaining of vimentin in Pan02 S.C. tumor of *S100a4-Cre; Ext1^{f/f}; Lsl-tdTomato*, and control (*S100a4-Cre; Lsl-tdTomato*) mice. White arrowheads indicate vimentin-positive fibroblasts. Scale bar = 50 μm.
(TIF)

**S6 Fig. Immunostaining of F4/80 and CD8α in Pan02 S.C. pancreatic tumor and RNA expression of MC38 S.C. tumor and Pan02 S.C. tumor.** (A) Immunohistochemistry for F4/80 of Pan02 S.C. tumor of *S100a4-Cre; Ext1^{f/f}* and control mice (left). The number of F4/80-positive cells in tumors of *S100a4-Cre; Ext1^{f/f}* and control mice (right). Data represent mean ± SEM (N = 6 for each cohort, unpaired t-test). Scale bar = 50 μm. (B) Immunohistochemistry for CD8α in peri- and intra-tumor regions of Pan02 S.C. tumors of *S100a4-Cre; Ext1^{f/f}* and control mice (left). The number of CD8α-positive cells in peri- and intra-tumor regions of tumors of *S100a4-Cre; Ext1^{f/f}* and control mice (right). Data represent mean ± SEM (N = 6 for each cohort, Mann–Whitney test). Scale bar = 50 μm. (C) Quantitative real-time RT-PCR analysis of genes related to the immune microenvironment in MC38 S.C. tumors of *S100a4-Cre; Ext1^{f/f}* and control mice. Data represent mean ± SEM (N = 4 for each cohort, Mann–Whitney test, $^{*}P < 0.05$). (D) Quantitative real-time RT-PCR analysis of *Mmp7* and *Tgfβ1* in Pan02 S.C. tumors of *S100a4-Cre; Ext1^{f/f}* and control mice. Data represent mean ± SEM (N = 4 for each cohort, Mann–Whitney test [*Mmp7*] and unpaired t-test [*Tgfβ1*], $^{*}P < 0.05$).
(TIF)

**S7 Fig. Double immunofluorescence of Ext1 and vimentin in human cancer and immunostaining of αSMA, CD163, and CD8α in human colon cancer and pancreatic cancer.** (A) Double immunofluorescence of Ext1 and vimentin in human cancer stroma. White

arrowheads indicate Ext1 and vimentin double-positive fibroblasts. Scale bar = 50 μm. (B) Immunostaining of αSMA in human cancer stroma (*top*). Arrowheads indicate that the vascular smooth muscle showed αSMA-positive as a positive control. Scale bar = 50 μm. Expression level of αSMA in human colon and pancreatic cancers of each Ext1 score (N = 3 each score) (bottom). (C) Immunostaining of CD163 and CD8α in human cancer stroma (left). Scale bar = 50 μm. The number of CD163- and CD8α-positive cells in human colon and pancreatic cancers of each Ext1 score (N = 3 each score) (right). Data represent mean ± SEM (N = 3 for each cohort, Mann–Whitney test, $^*P < 0.05$).
(TIF)

**S1 Table. Probe name of microarray data.**
(DOCX)

**S2 Table. Primer list of real-time RT-PCR data.**
(DOCX)

## Acknowledgments

We thank Kyoko Takahashi, Ayako Suga, Reiko Kitazumi, and Masayoshi Shimizu for their technical support. We also thank Editage (www.editage.com) for English language editing.

## Author Contributions

**Conceptualization:** Hiroyuki Tomita.

**Data curation:** Ayumi Niwa, Toshiaki Taniguchi, Hiroyuki Tomita.

**Formal analysis:** Ayumi Niwa.

**Funding acquisition:** Hiroyuki Tomita.

**Investigation:** Ayumi Niwa, Toshiaki Taniguchi, Hiroyuki Tomita, Takamasa Kinoshita, Chika Mizutani, Mikiko Matsuo, Yuko Imaizumi, Takahito Kuroda, Koki Ichihashi, Takaaki Sugiyama, Tomohiro Kanayama.

**Methodology:** Hiroyuki Tomita, Yu Yamaguchi.

**Project administration:** Hiroyuki Tomita.

**Resources:** Hiroyuki Tomita, Chika Mizutani.

**Software:** Hiroyuki Tomita.

**Supervision:** Hiroyuki Tomita, Hideshi Okada, Yu Yamaguchi, Shigeyuki Sugie, Nobuhisa Matsuhashi, Akira Hara.

**Validation:** Ayumi Niwa, Hiroyuki Tomita.

**Visualization:** Ayumi Niwa, Hiroyuki Tomita.

**Writing – original draft:** Ayumi Niwa.

**Writing – review & editing:** Hiroyuki Tomita.

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
