## [Decision Letter · Decision Letter 0]

16 Aug 2022

PONE-D-22-20421Conditional ablation of heparan sulfate expression in stromal fibroblasts promotes tumor growth in vivoPLOS ONE

Dear Dr. Tomita

Thank you for submitting your manuscript to PLOS ONE. After careful consideration, we feel that it has merit but does not fully meet PLOS ONE’s publication criteria as it currently stands. Therefore, we invite you to submit a revised version of the manuscript that addresses the points raised during the review process.

Reviewer 1 feels that the study is potentially interesting, but has several minor concerns that need to be addressed before publication. In contrast, reviewer 2 has critical concerns about the authors’ experimental conditions, including immunohistochemistry and statistical analysis (SD vs. SE). Reviewer 2 also requests the authors to change several images in Figures to more representative ones.

We look forward to receiving your revised manuscript.

Kind regards,

Hiroyasu Nakano, M.D., Ph.D.

Academic Editor

PLOS ONE

Journal Requirements:

3. To comply with PLOS ONE submissions requirements, in your Methods section, please provide additional information regarding the experiments involving animals and ensure you have included details on (1) methods of sacrifice, (2) methods of anesthesia and/or analgesia, and (3) efforts to alleviate suffering.

Reviewers' comments:

Reviewer's Responses to Questions

**Comments to the Author**

1. Is the manuscript technically sound, and do the data support the conclusions?

Reviewer #1: Yes

Reviewer #2: Partly

2. Has the statistical analysis been performed appropriately and rigorously? 

Reviewer #1: Yes

Reviewer #2: I Don't Know

3. Have the authors made all data underlying the findings in their manuscript fully available?

Reviewer #1: Yes

Reviewer #2: Yes

4. Is the manuscript presented in an intelligible fashion and written in standard English?

Reviewer #1: Yes

Reviewer #2: Yes

5. Review Comments to the Author

Reviewer #1: The manuscript by Niwa et al. has tried to tackle an important issue in cancer biology, which was the role of heparan sulfate (HS) in tumor progression. Given that one of the main sources of HS production in the tumor milieu is fibroblasts, the authors generated a mouse line in which the Ext1 gene was specifically depleted from S100a4-positive fibroblasts. The subcutaneous tumor transplantation experiments showed that HS deficiency in fibroblasts promoted tumor growth in both MC38 and Pan02 models, suggesting that HS may have a tumor-restraining role in this tumor model, although not conclusive at present. Interestingly, the infiltration of SMA-positive myofibroblasts in both the peritumoral and intratumoral regions was significantly affected in MC38 tumors developed in Ext1 conditional knockout mice, being accompanied with a decrease in the number of F4/80-postive macrophages. These findings were not consistently observed in the Pan02 tumor model, and this suggest differential effects of fibroblasts on tumor progression that depend on cancer types. Finally, the authors performed immunohistochemistry for Ext1 on tissue sections from human colon and pancreatic cancer, which showed the relevance of Ext1 expression in those human cancers. Collectively, I found that the authors have an interesting study, clearly addressing the role of fibroblast-derived HS in tumor progression, which will give some perspectives or insights on how HS production should be controlled in clinical settings to give benefit to patients with cancer. Some minor issues need to be addressed before publication.

Minor issues:

1) There seem to be some residual Ext1-positive cells in the skin of deltaS100a4 mice, and this suggested the possibility that some vessel cells or immune cells are expressing Ext1. It will be appreciated if the authors provide their speculation on this issue, citing previous articles in the literature.

2) I found one of the most striking data in the study was the significant differences in the infiltration or accumulation of SMA-positive fibroblasts or CAFs between MC38 and Pan02 tumors developed in control and deltaS100a4 mice (fig. 3A, 5A). It seemed to be that fibroinflammatory reaction or tissue repairing process was affected in the tumor tissues of the deltaS100a4 mice. Was this difference in SMA-positive fibroblast accumulation associated with the deposition of mucopolysaccharides or glycoproteins that can be detected by Alcian blue staining? If there is a positive correlation, HS production in fibroblast may be crucial for their infiltration in the tumor tissues.

3) The last sentence of the Abstract section "... suggesting that this effect is mediated ..." was unclear to this reviewer. I suggest the authors revise it so that it can be appreciated by readers.

Reviewer #2: Heparin sulfate (HS) is a glycocalyx component in the extracellular matrix and on HS proteoglycans. Heparan sulfate biosynthesis is catalyzed by the bi-functional glycosyltransferases Ext1 and Ext2. In this study, the authors examined the function of Ext1 in S100a4-expressing cells (Fibroblasts and so on) in cancer development.

The authors showed that subcutaneous transplantation of murine MC38 colon cancer cells and Pan02 pancreatic cancer cells in S100a4-Cre; Ext1flox/flox (dS100a4) mice showed enhanced tumor development compared to the control S100a4-Cre; tdTomato mice. MC38- transplanted dS100a4 mice showed decreased-aSMA-positive cells and F4/80+ cells in the tumor compared to control mice. In addition, Pan02-transplanted dS100a4 mice showed increased MMP7 expression in the tumor compared to control mice.

The authors showed that Ext1 expressed by S100a4+ cells regulate tumor progression in the subcutaneous tumor transplantation models. However, the exact mechanism regulating tumor development remains unclear because the authors only showed the correlation of the results.

In addition, the analysis methods and provided data need to be improved.

The specific comments are listed below.

1. The authors used S100a4-Cre mice. Since Ext1 is lost in S100a4-expressing cells throughout the body, it is uncertain whether the phenotype can be explained by local Ext1 expression in the tumors. For example, Are there differences in immune cell activation in lymph nodes or a spleen between dS100a4 and control mice before cancer transplantation? Other reports indicate that S100a4 is also expressed in immune cells (PMID: 30127784, PMID: 31775048, PMID: 34145030). There are possibilities that Ext1 regulates the activation of the leukocytes peripherally. The authors need to discuss these points.

2. In Figure 1A, Ext1 signals are also observed in Tomato-negative cells. Are these signals specific Ext1 signals? The author should show whether Ext1 protein levels are significantly decreased in dS100a4 mice in vivo by western blotting. In addition, Tomato positive signals also look like auto fluorescent signals. The authors need to show whether tomato-expressing cells and Ext1 expressing cells are fibroblasts by examing the expression of fibroblast-specific markers. Several reports showed that S100A4 was produced by macrophages, dendritic cells, or lymphocytes in vivo.

3. In Figure 1B, Tomato positive signals look like auto fluorescent signals. The authors need to change 1B to more representative data. In addition, the authors used mouse anti-HS antibodies with anti-mouse Ig H&L-Alexa488 antibodies. However, the authors examined mouse tissues, and there is a possibility that they are looking at the intrinsic IgG signal that mice originally had. Did they use specific mouse IgG blocking reagents? The reviewer thinks the authors need to use biotin- or fluorescent-labeled anti-HS antibodies for this analysis.

4. In Figures 2C and D, the authors discussed cell morphologies. However, it is not easy to understand the differences in morphologies from the provided pictures. The authors need to show enlarged images.

5. In Figures 3 and 5, the authors need to show whether HS is decreased in the tumor of dS100a4 mice.

6. In Figures 3 and 5, the authors need to show whether tomato+ cells express Ext1 in the tumor.

7. In Figures 3B and 5B, the authors need to show whether tomato+ cells were fibroblasts in the tumor of dS100a4 mice using fibroblast-specific markers.

8. In Figure 3C, the authors need to show whether they removed dead cells from the analysis.

9. In Figures 4C and 5D, the signal value data of the microarrays are not quantitative. The authors need to show the data of qPCR.

10. In Figure 5, the authors need to show whether CD8+ cells and F4/80+ cells were decreased or not in the tumor of dS100a4 mice.

11. In Figures 3 and 5, the authors showed that populations of a-SMA+ cells were decreased in the tumor of dS100a4 mice. It is known that HS binds to the TGF-b cytokine superfamily (PMID: 16709187, PMID: 28468283), and the levels of TGF-bs are tightly associated with the activation of a-SMA+ fibroblasts. The authors need to show whether the protein and mRNA levels of TGF-bs are decreased or not in the tumor of dS100a4 mice.

12. In Figure 6, the authors need to show whether Ext1+ cells were fibroblasts in the tumor using fibroblast-specific markers. It is not easy whether Ext1 expressing cells are fibroblasts are not from the provided data. In addition, it seems that epithelial cells or cancer cells also expressed Ext1 in Figure 6A.

13. In Figure 6, the authors need to show whether there is any correlation between Ext1 expression levels and the numbers of aSMA+ cells.

14. In Figure 6, the authors need to show whether there is any correlation between Ext1 expression levels and the numbers of F4/80+ cells.

15. In Figure 6, the authors need to show whether there is any correlation between Ext1 expression levels and the numbers of CD8+ cells.

16. It has already been reported that aSMA+ CAFs negatively regulate cancer development (PMID: 34108617, PMID: 24856586). The authors should discuss the correlation between these papers and their findings.

17. The authors used mixed standard deviations (SD) and standard errors (SE) in vivo mouse data. However, the authors examined the differences between the means of the two groups, so the author should use SE in vivo data.

18. The expression "dS100a4 mice" describes a mouse conditionally deficient in S100a4. The authors should change the words to like S100A4-Cre Ext1 f/f. (PMID: 31265437, PMID: 21911392)

19. In this paper, the authors examined only Ext1 expression in vivo. The authors also need to discuss the Ext2 expression levels in cancers.

20. In Material and Methods and Figure legends, the authors should show how the authors calculated image data and how many samples the authors examined.

21. In Material and Methods, the authors should show the catalog numbers of antibodies.

22. The reviewer thinks S100a4+ cells are not needed to be fibroblasts in this paper. This paper showed that Ext1 expressed by S100a4+ cells (maybe fibroblasts and immune cells) negatively regulate tumor development by activating aSMA+ myofibroblasts.

6. PLOS authors have the option to publish the peer review history of their article (what does this mean?). If published, this will include your full peer review and any attached files.

Reviewer #1: No

Reviewer #2: No

---

## [Author Response · Author response to Decision Letter 0]

15 Dec 2022

Response letter

Author responses to reviewers

Reviewer #1:

1) There seem to be some residual Ext1-positive cells in the skin of deltaS100a4 mice, and this suggested the possibility that some vessel cells or immune cells are expressing Ext1. It will be appreciated if the authors provide their speculation on this issue, citing previous articles in the literature.

Response: We thank you for your efforts in reviewing our manuscript. Some previous literature has deleted Ext1 in endothelial cells and observed the changes in endothelial cell function. Xingfeng Bao et al. (PMID: 21093315) reported that mutant mice with Ext1 deficiency in endothelial cells displayed a severe impairment in lymphocyte homing and decreased immune response. Kinoshita T et al. (PMID: 34790962) reported that endothelial cell-specific loss of Ext1 suppressed glioma growth in mice. Additionally, a previous study reported that there was a slight change in the number of developing B cells in B-cell Ext1-deficient mutant mice (PMID: 18479348). Furthermore, one study reported that Ext1 expression in multiple myeloma cells was a poor prognostic factor in multiple myeloma (PMID: 19298595). These reports suggest that Ext1 is expressed in immune cells in the stroma. We added the above context (page 15, lines 11–14) and above references as #29–#32 to the revised manuscript. 

2) I found one of the most striking data in the study was the significant differences in the infiltration or accumulation of SMA-positive fibroblasts or CAFs between MC38 and Pan02 tumors developed in control and deltaS100a4 mice (fig. 3A, 5A). It seemed to be that fibroinflammatory reaction or tissue repairing process was affected in the tumor tissues of the deltaS100a4 mice. Was this difference in SMA-positive fibroblast accumulation associated with the deposition of mucopolysaccharides or glycoproteins that can be detected by Alcian blue staining? If there is a positive correlation, HS production in fibroblast may be crucial for their infiltration in the tumor tissues.

Response: We thank you for your kind remarks. We performed Alcian blue staining on the MC38 and Pan02 tumors. In the MC38 tumor, decreased staining was observed in the peri-tumor region of the S100a4-Cre; Ext1f/f mice (Figure 3A). However, in the Pan02 tumor, no apparent differences in Alcian blue staining were observed between the S100a4-Cre; Ext1f/f and control mice (Figure 5A).

As for MC38 tumor, there was a decrease in fibroblasts and myofibroblasts in the S100a4-Cre; Ext1f/f mice (Figure 3B), as well as a decrease in HS production in the MC38 tumor (Figure S4A). Pan02 tumor showed decreased HS production in the tumor of the S100a4-Cre; Ext1f/f mice (Figure S5A), but unlike MC38 tumor, no significant difference was observed in the amount of fibroblasts or intratumoral myofibroblasts in the tumor of the S100a4-Cre; Ext1f/f mice (Figures 5B and 5C); it is suggested that in colon cancer, fibroblast’s own HS production affects their infiltration, but pancreatic cancer has less of an effect. We revised the manuscript (page 19, lines 17–19; page 22, lines 8–10; and page 25, lines 17–18) and added the Alcian blue staining image (Figures 3A and 5A).

3) The last sentence of the Abstract section "... suggesting that this effect is mediated..." was unclear to this reviewer. I suggest the authors revise it so that it can be appreciated by readers.

Response: Thank you for your kind comments. We revised it to “... by affecting the function and properties of cancer-associated fibroblasts, macrophages, and cancer cells” (Abstract, lines 18–19). 

Reviewer #2: 

 1. The authors used S100a4-Cre mice. Since Ext1 is lost in S100a4-expressing cells throughout the body, it is uncertain whether the phenotype can be explained by local Ext1 expression in the tumors. For example, Are there differences in immune cell activation in lymph nodes or a spleen between dS100a4 and control mice before cancer transplantation? Other reports indicate that S100a4 is also expressed in immune cells (PMID: 30127784, PMID: 31775048, PMID: 34145030). There are possibilities that Ext1 regulates the activation of the leukocytes peripherally. The authors need to discuss these points.

Response: We thank you for your efforts in reviewing our manuscript. We performed histopathological analysis of lymph nodes, spleen, and subcutaneous tissue to observe if there were histological differences in genotype. In the lymph nodes, both control and S100a4-Cre; Ext1f/f mice showed clear formation of lymph follicles and no apparent changes in tissue architecture. Additionally, no apparent changes were observed in the spleen. In the subcutaneous tissue, there was no apparent difference in the infiltrating immune cells (Figure S1C). We have added the above sentences (page 17, lines 6–12) to the revised manuscript. The above images (Figure S1C) were added to the manuscript.

2. In Figure 1A, Ext1 signals are also observed in Tomato-negative cells. Are these signals specific Ext1 signals? The author should show whether Ext1 protein levels are significantly decreased in dS100a4 mice in vivo by western blotting. In addition, Tomato positive signals also look like auto fluorescent signals. The authors need to show whether tomato-expressing cells and Ext1 expressing cells are fibroblasts by examing the expression of fibroblast-specific markers. Several reports showed that S100A4 was produced by macrophages, dendritic cells, or lymphocytes in vivo.

Response: We thank you for your kind remarks. We had tried western blotting, but the results were unstable due to problems with the quality of the antibody and other factors, making western blotting difficult. Instead, we added enlarged images of Ext1 protein expression of Tomato-positive fibroblasts in Figures 1A, so please review the images. These images can show reduced Ext1 protein expression in Tomato-positive fibroblasts of S100a4-Cre; Ext1f/f; lsl-tdTomato mice compared with control mice clearly. We performed vimentin staining using control (S100a4-Cre; lsl-tdTomato) mice and S100a4-Cre; Ext1f/f, lsl-tdTomato mice, and vimentin expression was confirmed in tomato-positive fibroblasts in each genotype (Figure S3A). Additionally, we performed double immunofluorescence for Ext1 and vimentin using control (Ext1f/f mice) and S100a4-Cre; Ext1f/f mice and confirmed vimentin expression in Ext1-positive fibroblasts of control mice (Figure S3B). We have added the results to the manuscript (page 17, lines 1–6) and Figure S3. 

3. In Figure 1B, Tomato positive signals look like auto fluorescent signals. The authors need to change 1B to more representative data. In addition, the authors used mouse anti-HS antibodies with anti-mouse Ig H&L-Alexa488 antibodies. However, the authors examined mouse tissues, and there is a possibility that they are looking at the intrinsic IgG signal that mice originally had. Did they use specific mouse IgG blocking reagents? The reviewer thinks the authors need to use biotin- or fluorescent-labeled anti-HS antibodies for this analysis.

Response: Thank you for your kind comments. HS staining of the subcutaneous tissue was performed again and Figure 1B was replaced (Figure 1B). Additionally, the staining was also performed with specific mouse IgG blocking reagents—Histofine MOUSESTAIN Kit (Nichirei, Tokyo, Japan). The above information has been added to the manuscript (page 10, lines 12–13).

4. In Figures 2C and D, the authors discussed cell morphologies. However, it is not easy to understand the differences in morphologies from the provided pictures. The authors need to show enlarged images.

Response: Thank you for your kind comments. We added enlarged images of the HE image in Figures 2C and D.

5. In Figures 3 and 5, the authors need to show whether HS is decreased in the tumor of dS100a4 mice.

Response: We thank you for your kind remarks. We performed HS immunostaining on MC38 tumors and Pan02 tumors. Both tumors showed decreased HS expression in S100a4-Cre; Ext1f/f mice (Figures S4A and S5A). We have added these results to the manuscript (page 21, lines 20-23; page 26, lines 10–13).

6. In Figures 3 and 5, the authors need to show whether tomato+ cells express Ext1 in the tumor.

Response: Thank you for your kind comments. We performed Ext1 immunostaining in MC38 and Pan02 tumors. In both tumors, Tomato-positive fibroblasts of control mice were Ext1-positive (Figures S4B and S5B). We have added these results to the manuscript (page 22, lines 1–4; page 26, lines 15-18).

7. In Figures 3B and 5B, the authors need to show whether tomato+ cells were fibroblasts in the tumor of dS100a4 mice using fibroblast-specific markers.

Response: Thank you for your kind comments. We performed vimentin immunostaining in MC38 and Pan02 tumors. In both tumors, Tomato-positive fibroblasts were vimentin-positive (Figures S4C and S5C). We have added these results to the manuscript (page 22, lines 1–5; and page 26, lines 15–19).

8. In Figure 3C, the authors need to show whether they removed dead cells from the analysis.

Response: Thank you for your kind comments. We performed flow cytometry analysis on fresh tumor tissues removing dead cells with SYTOX™ Red Dead Cell Stain (Invitrogen, USA). The results showed that the percentage of Tomato-positive fibroblasts in the tumor tissue of S100a4-Cre; Ext1f/f mice was significantly lower than that of control mice (Figure 3C). Since there were few dead cells in the peri-tumor region, original data of flow cytometry analysis of peri-tumor region has been included in Figure S2A. We have revised the sentences in the Materials and methods (page 12, lines 10–13) and Results (page 20, lines 3–5) sections according to the Figures 3C and S2A.

9. In Figures 4C and 5D, the signal value data of the microarrays are not quantitative. The authors need to show the data of qPCR.

Response: Thank you for your kind comments. We performed qPCR, and some mRNAs showed decreased expression in S100a4-Cre; Ext1f/f mice with significant differences (Figures S6C and S6D). We have added the results to the manuscript (page 24, lines 4–7 and lines 16–18; page 27, lines 3-4) with Figure S6C, D. 

10. In Figure 5, the authors need to show whether CD8+ cells and F4/80+ cells were decreased or not in the tumor of dS100a4 mice.

Response: Thank you for your kind comments. CD8a and F4/80 immunostaining showed no significant difference in the number of CD8+ cells and F4/80+ cells in each genotype of Pan 02 tumor (Figures S6A and S6B). We have added the results to the manuscript (page 28, lines 20–23) with Figures S6A and S6B. 

11. In Figures 3 and 5, the authors showed that populations of a-SMA+ cells were decreased in the tumor of dS100a4 mice. It is known that HS binds to the TGF-b cytokine superfamily (PMID: 16709187, PMID: 28468283), and the levels of TGF-bs are tightly associated with the activation of a-SMA+ fibroblasts. The authors need to show whether the protein and mRNA levels of TGF-bs are decreased or not in the tumor of dS100a4 mice.

Response: Thank you for your kind comments. We performed qPCR to investigate TGF-b1 expression. In MC38 tumors, there was a trend toward decreased expression of TGF-b1 in S100a4-Cre; Ext1f/f mice, but the difference was not significant (Figure S6C). In Pan02 tumor, TGF-b1 was significantly increased in S100a4-Cre; Ext1f/f mice (Figure S6D). Both MC38 and Pan02 showed decreased HS expression in the tumor of S100a4-Cre; Ext1f/f mice, but differences in TGF-b1 expression by genotype were not similar in both tumors. While there was a decrease of myofibroblasts in intratumoral region of MC38 tumor of S100a4-Cre; Ext1f/f mice, there was no significant difference in Pan02, which may also be related to the difference in TGF-b1 expression between MC38 and Pan02 tumors. We have added the above context to the Results section (page 22, lines 16–18; page 29, lines 3–5) with Figures S6C and S6D. 

12. In Figure 6, the authors need to show whether Ext1+ cells were fibroblasts in the tumor using fibroblast-specific markers. It is not easy whether Ext1 expressing cells are fibroblasts are not from the provided data. In addition, it seems that epithelial cells or cancer cells also expressed Ext1 in Figure 6A.

Response: Thank you for your kind comments. We performed double immunofluorescence of Ext1 and vimentin on human cancer stroma, and confirmed that Ext1-positive fibroblasts were vimentin-positive (Figure S7A). We have added the results to the manuscript (page 29, lines 22–24) with Figure S7A. 

13. In Figure 6, the authors need to show whether there is any correlation between Ext1 expression levels and the numbers of aSMA+ cells. 

Response: Thank you for your kind comments. For evaluating the expression intensity of aSMA, because it was difficult to evaluate by number, the expression intensity was divided into three levels: low, middle, and high. In human colon cancer, score-0 cases (N = 3) comprised 1 low and 2 middle intensity cases, score-1 cases (N = 3) comprised 2 middle and 1 high intensity cases, and score-2 cases (N = 3) comprised 1 middle and 2 high intensity cases (Figure S7B). In pancreatic cancer, score-0 cases (N = 3) comprised 1 low, 1 middle, and 1 high intensity cases; score-1 cases (N = 3) comprised 2 middle and 1 high intensity cases; and score-2 cases (N = 3) comprised 2 middle and 1 high intensity cases (Figure S7B). Colon cancers with higher Ext1 scores tended to have higher aSMA staining intensity. In pancreatic cancer, low intensity was observed only in score-0 cases, but there was no difference in aSMA staining intensity between score-1 and score-2 cases. Although further evaluation is needed due to the small number of cases, a correlation between Ext1 and aSMA expression was observed more in colon cancer than in pancreatic cancer. We have revised the manuscript (page 30, lines 12–20) and added the data (Figure S7B).

14. In Figure 6, the authors need to show whether there is any correlation between Ext1 expression levels and the numbers of F4/80+ cells.

Response: Thank you for your kind comments. We performed immunostaining for CD163, a marker of macrophages in humans, instead of F4/80. No significant differences were observed in the number of CD163-positive cells in both colon and pancreatic cancers (Figure S7C). We have revised the manuscript (page 30, lines 22–24; page 31, lines 1–2) and added the data (Figure S7C).

15. In Figure 6, the authors need to show whether there is any correlation between Ext1 expression levels and the numbers of CD8+ cells.

Response: Thank you for your kind comments. We performed immunostaining for CD8a and evaluated the number of CD8a-positive cells in colon and pancreatic cancers. There was no significant difference in the number of CD8a-positive cells among the different scores for colon cancer. In pancreatic cancer, the number of CD8a-positive cells tended to be higher in score-1 and score-2 cases than in score-0 cases, and a significant difference was observed between score-0 and score-1 cases. We have revised the manuscript (page 30, lines 22–24; page 31, lines 1–6) and added the data (Figure S7C).

16. It has already been reported that aSMA+ CAFs negatively regulate cancer development (PMID: 34108617, PMID: 24856586). The authors should discuss the correlation between these papers and their findings.

Response: Thank you for your kind comments. Studies (PMID: 34108617, PMID: 24856586) showed that αSMA+ CAF depletion led to the increase in colorectal or pancreatic cancer progression using αSMA-tk transgenic mice, leading to specific ablation of proliferating αSMA+ cells. Our experiments also showed a decrease of αSMA+ CAF in MC38 colon tumors and Pan02 pancreatic tumors. 

The paper (PMID: 34108617) showed that ablation of αSMA+ CAF in colorectal cancer leads to immunosuppression, including decreased CD8-positive T cells and increased Tregs. Although the difference was not significant, our study also showed that the number of CD8α-positive cells tended to decrease in S100a4-Cre; Ext1f/f mice, and the decreased number of macrophages and the expression of some cytokines such as Tnf suggested that immunosuppression occurred in the S100a4-Cre; Ext1f/f mice. The paper also showed that depletion of f αSMA+ CAF increased the Tgfβ1 secretion from the stromal cells. We observed downregulation of the TGF-β signaling pathway in tumors of S100a4-Cre; Ext1f/f mice in our study. Analysis of TGF-β expression throughout the tumor in our study may have made a difference. 

The study (PMID: 24856586) showed that pancreatic cancer with αSMA+ CAF depletion showed a different immune milieu, with decreased CD8-positive T cells and increased Tregs. In our experiment, no significant difference was observed in CD8α-positive T cells between S100a4-Cre; Ext1f/f and control mice. 

These differences may be because αSMA+ CAFs are not completely eliminated in our mouse model and that we analyzed tumors implanted subcutaneously. However, it can be said that the impact of αSMA+ CAF on tumor growth is significant. 

We added the above discussion (page 33, lines 23–24; page 34, lines 1–20) to the revised the manuscript.

17. The authors used mixed standard deviations (SD) and standard errors (SE) in vivo mouse data. However, the authors examined the differences between the means of the two groups, so the author should use SE in vivo data.

Response: Thank you for pointing out this issue. We have corrected the vivo data using SEM instead of SD and revised the manuscript.

18. The expression "dS100a4 mice" describes a mouse conditionally deficient in S100a4. The authors should change the words to like S100A4-Cre Ext1 f/f. (PMID: 31265437, PMID: 21911392)

Response: Thank you for pointing out this important issue. We have revised the manuscript and figures using S100a4-Cre Ext1 f/f instead of dS100a4.

19. In this paper, the authors examined only Ext1 expression in vivo. The authors also need to discuss the Ext2 expression levels in cancers.

Response: Thank you for your kind comments. Germline heterozygous loss-of-function in Ext1 or Ext2 is a known cause of multiple exostoses (HME) (PMID: 29277722, PMID: 24128412). Ext1 and Ext2 mutations have been reported in tumors other than HME. In leukemia and non-melanoma skin cancer cells, epigenetic inactivation of EXT1 by promoter hypermethylation is found and the tumor suppressing effect of Ext1 was revealed (PMID: 15385438). However, in multiple myeloma, the correlation of high expression of EXT1 and a poor prognosis was reported (PMID: 19298595). In another previous paper, EXT2 mutation in breast carcinoma patients was reported (PMID: 22205677). Furthermore, Drake et al. (PMID: 22309216) reported that both EXT1 and EXT2 have altered N-glycosylation in human aggressive breast cancer cells. In our study, the TME with Ext1 loss in fibroblasts proliferates colon and pancreatic cancer growth. Although our study did not analyze the expression of Ext1 and Ext2 in cancers in detail, it is suggested that changes in the expression of Ext1 or Ext2 in cancers may contribute to the differences in cancer development in this study. Further analysis of the expression of Ext1 and Ext2 in cancer cells and fibroblasts is needed to gain a better understanding of how TME with Ext1 loss of fibroblasts can lead to cancer development. We have added the discussion to the manuscript (page 35, lines 16–24; page 36, lines 1–6).

20. In Material and Methods and Figure legends, the authors should show how the authors calculated image data and how many samples the authors examined.

Response: Thank you for your kind comments. Positive cell count of immunohistochemistry was performed using the cell counting command of NIH Image J software. Measurement of the thickness of the αSMA (+) fibroblasts layer was also performed with Image J. Additionally, the fluorescent images were transformed to 8-bit type and thresholds were set, and then, the area of Tomato-positive cells, the area of Tomato (+) αSMA (+) fibroblasts, and the fluorescence intensity value of HS were measured with Image J. We have revised the manuscript and the above methods (page 13, lines 19–21; page 14, lines 1–4) and the number of samples used were shown in the manuscript and figure legends.

21. In Material and Methods, the authors should show the catalog numbers of antibodies.

Response: Thank you for pointing out this. We have added the catalog numbers of antibodies.

22. The reviewer thinks S100a4+ cells are not needed to be fibroblasts in this paper. This paper showed that Ext1 expressed by S100a4+ cells (maybe fibroblasts and immune cells) negatively regulate tumor development by activating aSMA+ myofibroblasts.

Response: Thank you for your kind comments. In our study, we used S100a4 as a marker for fibroblasts and focused our discussion on the function of heparan sulfate in fibroblasts, but as you said, it would be more correct to view this study as looking at the function of the TME, where Ext1 expression in S100a4-positive cells is deleted. I thought that a further analysis of which cells in the stroma express S100a4 will be necessary in the future. We have added the related discussion to the manuscript (page 36, lines 13–16).

---

## [Decision Letter · Decision Letter 1]

24 Jan 2023

PONE-D-22-20421R1Conditional ablation of heparan sulfate expression in stromal fibroblasts promotes tumor growth in vivoPLOS ONE

Dear Dr. Tomita,

Thank you for submitting your manuscript to PLOS ONE. After careful consideration, we feel that it has merit but does not fully meet PLOS ONE’s publication criteria as it currently stands. Therefore, we invite you to submit a revised version of the manuscript that addresses the points raised during the review process.

We look forward to receiving your revised manuscript.

Kind regards,

Hiroyasu Nakano, M.D., Ph.D.

Academic Editor

PLOS ONE

Journal Requirements:

Reviewers' comments:

Reviewer's Responses to Questions

**Comments to the Author**

1. If the authors have adequately addressed your comments raised in a previous round of review and you feel that this manuscript is now acceptable for publication, you may indicate that here to bypass the “Comments to the Author” section, enter your conflict of interest statement in the “Confidential to Editor” section, and submit your "Accept" recommendation.

Reviewer #1: All comments have been addressed

Reviewer #2: All comments have been addressed

2. Is the manuscript technically sound, and do the data support the conclusions?

Reviewer #1: Yes

Reviewer #2: Partly

3. Has the statistical analysis been performed appropriately and rigorously? 

Reviewer #1: Yes

Reviewer #2: Yes

4. Have the authors made all data underlying the findings in their manuscript fully available?

Reviewer #1: Yes

Reviewer #2: Yes

5. Is the manuscript presented in an intelligible fashion and written in standard English?

Reviewer #1: Yes

Reviewer #2: Yes

6. Review Comments to the Author

Reviewer #1: The authors have addressed all of the three comments previously raised by this reviewer, which seemed to be all satisfactory. They truthfully showed that the dependence of tumor growth on HS might be different between tumor types, and this could be a useful information for us to consider inter-tumoral heterogeneity that is also relevant to human cases. I have no additional comments on the manuscript and recommend it for publication.

Reviewer #2: The revised manuscript has addressed many of my concerns, however, there are still a few issues that need to be addressed before publication.

1. In Supplementary 4C, the images of tdTomato in the control and S100a4-Cre Ext1 f/f groups appear to be identical and should be replaced with more appropriate images.

2. In Supplementary 5C, tdTomato and vimentin signals in S100a4-Cre Ext1 f/f are too perfectly matched, which is strange given that vimentin-negative cells also express S100a4, as shown in Figure 3A. More suitable images should be used.

3. In Supplementary 6C and D, it is unclear how the relative expression values were calculated and relative to what value. Could the authors provide more information on this? Also, the data should be separated by each gene, not combined in one figure.

7. PLOS authors have the option to publish the peer review history of their article (what does this mean?). If published, this will include your full peer review and any attached files.

Reviewer #1: No

Reviewer #2: No

---

## [Author Response · Author response to Decision Letter 1]

25 Jan 2023

Response letter

Author responses to reviewers

Reviewer #1: 

The authors have addressed all of the three comments previously raised by this reviewer, which seemed to be all satisfactory. They truthfully showed that the dependence of tumor growth on HS might be different between tumor types, and this could be a useful information for us to consider inter-tumoral heterogeneity that is also relevant to human cases. I have no additional comments on the manuscript and recommend it for publication.

Response: Thank you so much for your cooperation.

Reviewer #2: 

The revised manuscript has addressed many of my concerns, however, there are still a few issues that need to be addressed before publication.

1. In Supplementary 4C, the images of tdTomato in the control and S100a4-Cre Ext1 f/f groups appear to be identical and should be replaced with more appropriate images.

Response: Thank you for your indication. We have changed it with the appropriate image. Please see revised Supplementary Fig. 4C. 

2. In Supplementary 5C, tdTomato and vimentin signals in S100a4-Cre Ext1 f/f are too perfectly matched, which is strange given that vimentin-negative cells also express S100a4, as shown in Figure 3A. More suitable images should be used.

Response: Yes, we agree. We have changed and enlarged the figures to show vimentin-negative cells clearly. Please see revised Supplementary Fig. 5C. 

3. In Supplementary 6C and D, it is unclear how the relative expression values were calculated and relative to what value. Could the authors provide more information on this? Also, the data should be separated by each gene, not combined in one figure.

Response: Thank you for your suggestion. To analyze relative gene expression, the comparative Ct method was used and relative values of gene expression levels were calculated by setting the expression level of the sample with the lowest expression level as 1. We have revised the Materials and methods in the manuscript (page 13, lines 15-18). We also changed the data for each gene separately. Please see revised Supplementary Figure 6C and D.

---

## [Decision Letter · Decision Letter 2]

2 Feb 2023

Conditional ablation of heparan sulfate expression in stromal fibroblasts promotes tumor growth in vivo

PONE-D-22-20421R2

Dear Dr. Tomita,

We’re pleased to inform you that your manuscript has been judged scientifically suitable for publication and will be formally accepted for publication once it meets all outstanding technical requirements.

Kind regards,

Hiroyasu Nakano, M.D., Ph.D.

Academic Editor

PLOS ONE

Additional Editor Comments (optional):

Reviewers' comments:

Reviewer's Responses to Questions

**Comments to the Author**

1. If the authors have adequately addressed your comments raised in a previous round of review and you feel that this manuscript is now acceptable for publication, you may indicate that here to bypass the “Comments to the Author” section, enter your conflict of interest statement in the “Confidential to Editor” section, and submit your "Accept" recommendation.

Reviewer #2: All comments have been addressed

2. Is the manuscript technically sound, and do the data support the conclusions?

Reviewer #2: Yes

3. Has the statistical analysis been performed appropriately and rigorously? 

Reviewer #2: Yes

4. Have the authors made all data underlying the findings in their manuscript fully available?

Reviewer #2: Yes

5. Is the manuscript presented in an intelligible fashion and written in standard English?

Reviewer #2: Yes

6. Review Comments to the Author

Reviewer #2: (No Response)

7. PLOS authors have the option to publish the peer review history of their article (what does this mean?). If published, this will include your full peer review and any attached files.

Reviewer #2: No

---

## [Editor Report · Acceptance letter]

7 Feb 2023

PONE-D-22-20421R2 

Conditional ablation of heparan sulfate expression in stromal fibroblasts promotes tumor growth <i>in vivo<i> 

Dear Dr. Tomita:

I'm pleased to inform you that your manuscript has been deemed suitable for publication in PLOS ONE. Congratulations! Your manuscript is now with our production department. 

Kind regards, 

on behalf of

Professor Hiroyasu Nakano 

Academic Editor

PLOS ONE